# Continuous evaluation of denoising strategies in resting-state fMRI connectivity using fMRIPrep and Nilearn

**Hao-Ting Wang**[1]*, **Steven L. Meisler**[2,3], **Hanad Sharmarke**[1], **Natasha Clarke**[1], **Nicolas Gensollen**[4], **Christopher J. Markiewicz**[5], **François Paugam**[1,6,7], **Bertrand Thirion**[4], **Pierre Bellec**[1,8]

**1** Centre de recherche de l'institut Universitaire de gériatrie de Montréal (CRIUGM), Montréal, Québec, Canada, **2** Program in Speech and Hearing Bioscience and Technology, Harvard University, Massachusetts, United States of America, **3** Department of Brain and Cognitive Sciences, Massachusetts Institute of Technology, Massachusetts, United States of America, **4** Inria, CEA, Université Paris-Saclay, Paris, France, **5** Department of Psychology, Stanford University, Stanford, United States of America, **6** Computer Science and Operations Research Department, Université de Montréal, Montréal, Québec, Canada, **7** Mila—Institut Québécois d'Intelligence Artificielle, Montréal, Canada, **8** Psychology Department, Université de Montréal, Montréal, Québec, Canada

* wang.hao-ting@criugm.qc.ca

**Data Availability Statement:** Research code is available on GitHub repository (https://github.com/SIMEXP/fmriprep-denoise-benchmark). Datasets used in the current study are existing open access

## Abstract

Reducing contributions from non-neuronal sources is a crucial step in functional magnetic resonance imaging (fMRI) connectivity analyses. Many viable strategies for denoising fMRI are used in the literature, and practitioners rely on denoising benchmarks for guidance in the selection of an appropriate choice for their study. However, fMRI denoising software is an ever-evolving field, and the benchmarks can quickly become obsolete as the techniques or implementations change. In this work, we present a denoising benchmark featuring a range of denoising strategies, datasets and evaluation metrics for connectivity analyses, based on the popular fMRIprep software. The benchmark prototypes an implementation of a reproducible framework, where the provided Jupyter Book enables readers to reproduce or modify the figures on the Neurolibre reproducible preprint server (https://neurolibre.org/). We demonstrate how such a reproducible benchmark can be used for continuous evaluation of research software, by comparing two versions of the fMRIprep. Most of the benchmark results were consistent with prior literature. Scrubbing, a technique which excludes time points with excessive motion, combined with global signal regression, is generally effective at noise removal. Scrubbing was generally effective, but is incompatible with statistical analyses requiring the continuous sampling of brain signal, for which a simpler strategy, using motion parameters, average activity in select brain compartments, and global signal regression, is preferred. Importantly, we found that certain denoising strategies behave inconsistently across datasets and/or versions of fMRIPrep, or had a different behavior than in previously published benchmarks. This work will hopefully provide useful guidelines for the fMRIprep users community, and highlight the importance of continuous evaluation of research methods.

datasets on OpenNeuro (https://openneuro.org/datasets/ds000228/versions/1.1.0, https://openneuro.org/datasets/ds000030/versions/1.0.0). All metadata and summary statistics are available on Zenodo (https://doi.org/10.5281/zenodo.6941757). Retrieval of the data mentioned above are retrievable through the code repository and the Neurolibre preprint service (https://doi.org/10.55458/neurolibre.00012).

**Funding:** The project was supported by the following fundings: Digital Alliance Canada Resource Allocation Competition (RAC 1827 and RAC 4455; https://alliancecan.ca/) to PB, Institut de Valorisation des Données projets de recherche stratégiques (IVADO PFR3; https://ivado.ca/) to PB, and Canadian Consortium on Neurodegeneration in Aging (CCNA; team 9 "discovering new biomarkers"; https://ccna-ccnv.ca/) to PB, the Courtois Foundation to PB, and Institut national de recherche en sciences et technologies du numérique (INRIA; Programme Équipes Associées - NeuroMind Team DRI-012229; https://www.inria.fr/) to PB and BT. HTW and NC were funded by Institut de valorisation des données (IVADO) postdoctoral research funding. SLM was funded by the National Institute on Deafness and Other Communication Disorders (NIDCD; Grant 5T32DC000038; https://www.nidcd.nih.gov/). CJM was funded by the National Institute of Mental Health (NIMH, Grant 5R24MH117179; https://www.nimh.nih.gov/). FP was funded by Courtois Neuromod (https://www.cneuromod.ca/). PB was funded by Fonds de Recherche du Québec - Santé (FRQ-S; https://frq.gouv.qc.ca/en/). The sponsors or funders were not involved in the study design, data collection and analysis, decision to publish, or preparation of the manuscript.

**Competing interests:** The authors have declared that no competing interests exist.

## Author summary

Many strategies exist to denoise fMRI signal. However, denoising software is ever-evolving, and benchmarks quickly become obsolete. Here, we present a denoising benchmark featuring several strategies and datasets to evaluate functional connectivity analysis, based on fMRIprep. The benchmark is implemented in a fully reproducible framework. The provided Jupyter Book enables readers to reproduce core computations and figures from the Neurolibre reproducible preprint server (https://neurolibre.org/). Most results were consistent with prior literature. Scrubbing was generally effective, but is incompatible with statistical analyses requiring the continuous sampling of brain signals, for which a simpler strategy is preferred. Importantly, we found that certain denoising strategies behaved inconsistently across datasets and/or fMRIPrep versions, or differently from the literature. Our benchmark can enable the continuous evaluation of research software and provide up-to-date denoising guidelines to fMRIprep users. This generic reproducible infrastructure can facilitate the continuous evaluation of research tools across various fields.

## Introduction

Resting-state functional magnetic resonance imaging (fMRI) is a tool for studying human brain connectivity [1,2] which comes with many analytical challenges [3,4]. One such key challenge is the effective correction of non-neuronal sources of fluctuations (called confounds), known as denoising, which is important to reduce bias when studying the association between connectomes and behavioral measures of interest [5]. A wide range of denoising strategies have been proposed in the literature, with no approach emerging as a clear single best solution. Denoising benchmarks on functional connectivity [6,7] have thus become an important resource for the community to understand which denoising strategy is most appropriate in a given study. Denoising benchmarks are however at a constant risk of becoming obsolete, with new strategies being regularly developed or revised, as well as an ever-expanding scope of populations being enrolled in research studies. The main objective of the present work is to develop a fully reproducible denoising benchmark for fMRI functional connectivity, and demonstrate how the proposed infrastructure enables the continuous evaluation of denoising strategies across multiple software versions and datasets.

Reproducible and robust results have become a recurring interest in the neuroimaging community [8,9]. The popular package fMRIPrep [10] is a prominent solution for fMRI preprocessing designed with reproducibility in mind, and we decided to build upon that software for our benchmark. However, fMRIPrep only performs minimal preprocessing while generating a broad list of potential confounds, intentionally leaving the selection of the exact denoising strategy to end-users. The connectivity metrics are also not included as part of fMRIPrep outputs, and users rely on additional software to apply denoising to time series and generate connectivity measures. One popular open-source Python library for this purpose is Nilearn [11]. Yet, until recently, there was no straightforward way to incorporate fMRIPrep outputs into Nilearn in order to reproduce the most common denoising strategies. This lack of integration represented a major barrier to the exploration of denoising tools, both for cognitive neuroscientists who were required to develop custom code, and for computer scientists who had to develop a detailed understanding of the inner workings of denoising strategies and fMRIPrep.

The main references for denoising benchmarks [6,7] did not use the then-novel fMRIPrep. Whether the results of these benchmarks remain consistent with fMRIPrep outputs is an open question. Different fMRI preprocessing softwares provide largely similar results, but noticeable differences are still present [12,13]. Other computational factors can possibly impact the conclusion of a benchmark, such as the version of software and operating system [14]. Recent research has also demonstrated that, given one fMRI dataset and similar research goals, different researchers will select a wide variety of possible analytical paths [8]. The lack of standard integration between fMRIPrep and Nilearn could lead to differences (and errors) in the implementation of the same denoising strategies by researchers, which can in turn lead to marked differences in the impact of denoising methods.

In this work, we propose to address the issues of robustness of functional connectivity denoising benchmarks by building a fully reproducible solution. This reproducible benchmark will allow the fMRI research community to consolidate past knowledge on technical advances, examine computation instability across different software versions, and provide guidance for practitioners. For the broader scientific research community, we aim to highlight the importance of continuous method evaluation, and propose a widely applicable infrastructure to implement such benchmarks. In order to create this benchmark, we implemented a series of specific objectives:

- First, we developed a standardized application programming interface (API) to extract nuisance regressors from fMRIPrep. The robust API, which was added to Nilearn release 0.9.0, can be used to flexibly retrieve a subset of fMRIPrep confounds for denoising and precisely replicate nuisance regressors based on denoising strategies proposed in the literature.

- Our second objective was to implement a denoising benchmark to provide recommendations on the choice of functional connectivity denoising strategies for fMRIPrep users. We used easily fetchable open access data, specifically two datasets on OpenNeuro [15] with diverse participant profiles: *ds000228* [16] and *ds000030* [17]. *ds000228* contains adult and child samples, and *ds000030* includes psychiatric conditions. The benchmark systematically evaluates the impact of denoising choices using a series of metrics based on past research [6,7].

- Our third objective was to turn this benchmark into a fully reproducible and interactive research object. We combined a series of technologies, including software containers [18], the Jupyter Book project [19], and the NeuroLibre preprint service [20] in order to create the first fully reproducible benchmark of denoising strategies for fMRI resting-state connectivity.

- Our fourth and last objective was to demonstrate that our approach can be used to evaluate the robustness of the benchmark, by identifying possible differences across multiple versions of fMRIPrep.

## Results

### Software implementation

We designed two APIs for users to perform denoising of fMRI time series using Nilearn, based on fMRIPrep outputs. The APIs are maintainable, i.e., composed of modular and well-tested code, and user-friendly, i.e., the syntax is standard and robust to errors. The confounds are loaded by the APIs in a format compatible with downstream Nilearn analysis functions. The first, basic API retrieves different classes of confound regressors sorted in categories of noise,

`nilearn.interfaces.fmriprep.load_confounds` (simplified as `load_confounds` in the following sections). The second, higher level API implements common strategies from the denoising literature, `nilearn.interfaces.fmriprep.load_confounds_strategy` (simplified as `load_confounds_strategy` in the following sections). The `load_confounds` and `load_confounds_strategy` APIs are available from Nilearn version 0.9.0 onwards. The following section describes both APIs in greater detail.

**load_confounds: basic noise components.**   The following Python code snippet demonstrates the basic usage of `load_confounds`.

```
from nilearn.interfaces.fmriprep import load_confounds
confounds_simple, sample_mask = load_confounds(
  fmri_filenames,
  strategy = ["high_pass", "motion", "wm_csf"],
  motion = "basic", wm_csf = "basic"
)
```

- `fmri_filenames`: path to processed image files, optionally as a list of paths.

- `strategy`: A list defining the categories of confound variables to use. Amongst the three in this example, `motion` and `wm_csf` are further tunable.

- `motion` and `wm_csf`: additional parameters with four options

  o `basic`: original parameters

  o `power2`: original parameters + quadratic terms

  o `derivatives`: original parameters + 1st temporal derivatives

  o `full`: original parameters + 1st temporal derivatives + quadratic terms + power2d derivatives

The `load_confounds` API fetches specific categories of confound variables, such as motion parameters. It is possible to fine-tune these categories through various options, such as the order of expansion of motion parameters. The implementation only supports fMRIPrep version 1.4 and above, and requires the fMRIPrep output directory in its original format. Users specify the path of a preprocessed functional file (file ending with `desc-preproc_bold.nii.gz` or `desc-smoothAROMAnonaggr_bold.nii.gz` in the case of ICA-AROMA). Warnings and errors inform the user if files or confounds were missing, for example if fMRIPrep was run without the option for ICA-AROMA yet users request ICA-AROMA confounds, or try to load an preprocessed fMRI output not suited for combination with ICA-AROMA regressors. The function returns the confound variables in a Pandas `DataFrame` object [21,22] and a time sample mask. The sample mask indexes the time points to be kept. The function can also be used with a list of input files, in which case it returns a list of confounds `DataFrames` and a list of time sample masks. A parameter called `strategy` can be used to pass a list of different categories of noise regressors to include in the confounds: `motion, wm_csf, global_signal, scrub, compcor, ica_aroma, high_pass, non_steady_state`. For each noise category, additional function parameters are available to tune the corresponding noise variables (please refer to Nilearn documentation [23] for more details). Please refer to the following summary for the parameters listed above. See S1 Text Annex A for a literature review and discussion for each category of common noise sources.

- `motion`: head motion estimates. Associated parameter: motion

- `wm_csf` confounds derived from white matter and cerebrospinal fluid. Associated parameter: wm_csf

- `global_signal` confounds derived from the global signal. Associated parameter: `global_signal`

- `compcor` confounds derived from CompCor [1]. When using this noise component, `high_pass` must also be applied. Associated parameter: `compcor, n_compcor`

- `ica_aroma` confounds derived from ICA-AROMA [2]. Associated parameter: `ica_aroma`

- `scrub` regressors for [3] scrubbing approach. Associated parameter: `scrub, fd_threshold, std_dvars_threshold`

**load_confounds_strategy: pre-defined strategies.** The following code snippet demonstrates the basic usage of `load_confounds_strategy`. This snippet retrieves the same confounds variables as described in the example for `load_confounds`.

```
from nilearn.interfaces.fmriprep import load_confounds_strategy
confounds_simple, sample_mask = load_confounds_strategy(
fmri_filenames,
denoise_strategy = "simple")
```

- `fmri_filenames`: path to processed image files, optionally as a list of paths.

- `denoise_strategy`: The name of a predefined strategy (see Table 1).

`load_confounds_strategy` provides an interface to select a complete set of curated confounds reproducing a common strategy used in the literature, with limited parameters for

**Table 1. Correspondence of load_confounds parameters to predefined denoising strategies in load_confounds_strategy.**

| Parameters | Strategy | | | |
| --- | --- | --- | --- | --- |
|  | **simple** | **scrubbing** | **compcor** | **ica_aroma** |
| high_pass | True | True | True | True |
| motion | full* | full* | full* | N/A |
| wm_csf | basic* | full | N/A | basic* |
| global_signal | None* | None* | None*† | None* |
| scrub | N/A | 5* | N/A | N/A |
| fd_threshold | N/A | 0.2*^ | N/A | N/A |
| std_dvars_threshold | N/A | 3*^ | N/A | N/A |
| compcor | N/A | N/A | anat_combined* | N/A |
| n_compcor | N/A | N/A | all* | N/A |
| ica_aroma | N/A | N/A | N/A | full |
| demean | True* | True* | True* | True* |

* Parameters with customisable parameters.

^ The default thresholds will be updated in nilearn version 0.13.0 to match fMRIPrep defaults (fd_threshold = 0.5, std_dvars_threshold = 1.5). A deprecation warning has been added in version 0.10.3.

† In version 0.10.3, global_signal is an allowed parameter to reflect the documentation of fMRIPrep. The default remains as not applying global_signal.

user customisation. There are four possible strategies that can be implemented from fMRIPrep confounds:

- `simple` [24]: motion parameters and tissue signal

- `scrubbing` [25]: volume censoring, motion parameters, and tissue signal

- `compcor` [26]: anatomical compcor and motion parameters

- `ica_aroma` [27]: ICA-AROMA based denoising and tissue signal

All strategies, except `compcor`, provide an option to add global signal to the confound regressors. The predefined strategies and associated noise components are listed in Table 1. Parameters that can be customized are indicated with a *. See the Nilearn documentation for more details [28]. See S1 Text Annex B for a more in-depth review of common denoising strategies in the literature and S1 Text Annex C for a summary of evaluation benchmarks using these strategies.

**Denoising workflow.** The denoising workflow is implemented through Nilearn. Fig 1 presents the graphic summary of the workflow. An fMRI dataset in the Brain Imaging Data Structure (BIDS) standard was first passed to fMRIPrep. Brain parcellation atlases were retrieved through the TemplateFlow [29] Python client (see https://www.templateflow.org/usage/client/). In cases where an atlas was absent from TemplateFlow, it was converted into TemplateFlow naming convention to enable use of the Python client. Each atlas was passed to the `NiftiLabelsMasker` or `NiftiMapsMasker` for time series extraction. fMRIPrep outputs were input to a Nilearn-based connectome generating workflow using `load_confounds_strategy`. The filtered confounds and the corresponding preprocessed NIFTI images were then passed to the Nilearn masker generated with the atlas where the

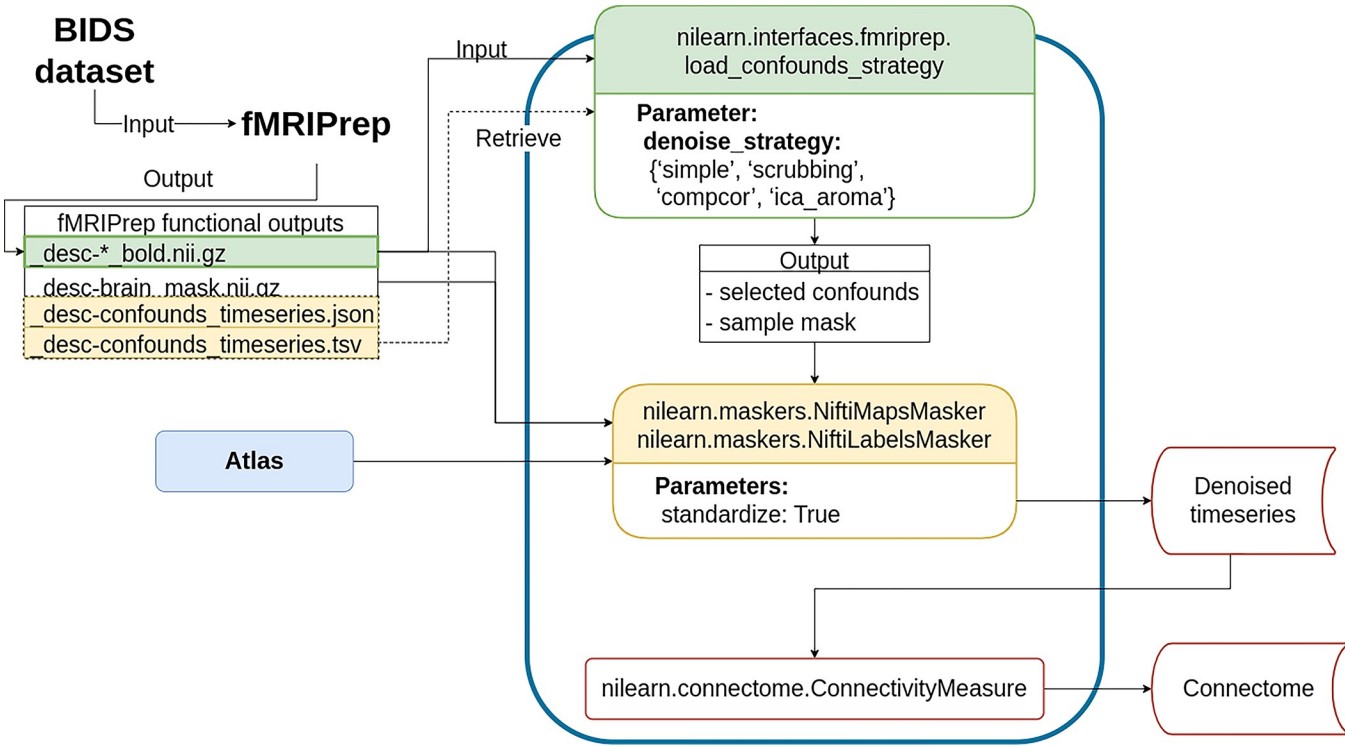

**Fig 1. Workflow for post-fMRIPrep time series extraction with Nilearn tools.**

underlying function nilearn.signals.clean applied the regressors for denoising (see https://nilearn.github.io/stable/modules/generated/nilearn.signal.clean.html). S1 Text Annex E contains the mathematical operation implemented by the denoising procedure. The time series and connectomes were saved as the main outputs for further analysis.

The Python-based workflow describes the basic procedure to generate functional connectomes from fMRIPrep outputs with a Nilearn data loading routine (e.g., `NiftiMapsMasker` or `NiftiLabelsMasker`), fMRIPrep confounds output retrieval function (e.g., `load_confounds_strategy`), and connectome generation routine (`ConnectivityMeasure`). Path to the preprocessed image data is passed to `load_confounds_strategy` and the function fetches the associated confounds from the `.tsv` file. The path of an atlas and the path of the preprocessed image file is then passed to the masker, along with the confounds, for time series extraction. The time series are then passed to `ConnectivityMeasure` for generating connectomes.

**Benchmark workflow.** OpenNeuro datasets were retrieved through DataLad [30] and fMRIPrep images were pulled from DockerHub. SLURM job submission scripts to process the fMRI data were generated with the Python tool fMRIPrep-SLURM (https://github.com/SIMEXP/fmriprep-slurm). The fMRIPrep derivatives and atlas retrieved from the Template-Flow archive were passed to the connectome workflow described in Fig 1. We extracted the signals using a range of atlases at various resolutions (see Materials and Methods for details). For each parcellation scheme and each fMRI dataset, 10 sets of time series were generated, including one baseline and 9 different denoising strategies (see Table 2). We report the quality metrics and break down the effect on each dataset, preprocessed with fMRIPrep 20.2.1 long-term support branch (LTS). Motion characteristics were also generated per dataset and used to exclude fMRI runs with excessive motion from entering the benchmark. Trends in each atlas were similar, so we combined all atlases for the following report. The detailed breakdown by parcellation scheme can be found in the associated Jupyter Book [31]. Fig 2 presents a graphical summary of the benchmark workflow.

## Benchmark results from fMRIPrep 20.2.1 LTS

We reported the demographic information and the gross mean framewise displacement before and after excluding subjects with high motion. We then aimed to assess the overall similarity between connectomes generated from each denoising strategy, and evaluated the denoising strategies using four metrics from Ciric and colleagues' benchmark [6]:

1. Loss of degrees of freedom: sum of number of regressors used and number of volumes censored.

2. Quality control / functional connectivity [QC-FC; 34]: partial correlation between motion and connectivity with age and sex as covariates.

3. Distance-dependent effects of motion on connectivity [DM-FC; 25]: correlation between node-wise Euclidean distance and QC-FC.

4. Network modularity [4]: graph community detection based on Louvain method, implemented in the Brain Connectome Toolbox.

**Significant differences in motion levels existed both between datasets, and within-dataset, across clinical and demographic subgroups.** We applied a motion threshold to exclude subjects with marked motion in the two OpenNeuro datasets: dataset *ds000228* (N = 155) [16] and dataset *ds000030* (N = 212) [17]. Table 3 shows the demographic information of subjects

**Table 2. Strategies examined in the benchmark and associated parameters applied to load_confounds.**

| strategy | image | high_pass | motion | wm_csf | global_signal | scrub | fd_thresh | std_dvars_threshold | compcor | n_compcor | ica_aroma |
|---|---|---|---|---|---|---|---|---|---|---|---|
| baseline | desc-preproc_bold | True | N/A | N/A | N/A | N/A | N/A | N/A | N/A | N/A | N/A |
| simple | desc-preproc_bold | True | full | basic | N/A | N/A | N/A | N/A | N/A | N/A | N/A |
| simple+gsr | desc-preproc_bold | True | full | basic | basic | N/A | N/A | N/A | N/A | N/A | N/A |
| scrubbing.5 | desc-preproc_bold | True | full | full | N/A | 5 | 0.5 | None | N/A | N/A | N/A |
| scrubbing.5+gsr | desc-preproc_bold | True | full | full | basic | 5 | 0.5 | None | N/A | N/A | N/A |
| scrubbing.2 | desc-preproc_bold | True | full | full | N/A | 5 | 0.2 | None | N/A | N/A | N/A |
| scrubbing.2+gsr | desc-preproc_bold | True | full | full | basic | 5 | 0.2 | None | N/A | N/A | N/A |
| compcor | desc-preproc_bold | True | full | N/A | N/A | N/A | N/A | N/A | anat_combined | all | N/A |
| compcor6 | desc-preproc_bold | True | full | N/A | N/A | N/A | N/A | N/A | anat_combined | 6 | N/A |
| aroma | desc-smoothAROMAnonaggr_bold | True | N/A | basic | N/A | N/A | N/A | N/A | N/A | N/A | full |

* 50% variance explained.

** In Ciric et al. (2017), there was a variation of the ICA-AROMA strategy including global signal regressor. The global signal regressor generated by fMRIPrep does not follow the recommendation of Pruim et al. [32]. The result of ICA-AROMA+GSR can be found in the first version of the preprint.

*** Referring to the non-aggressive implementation in Pruim and colleagues' work [32]

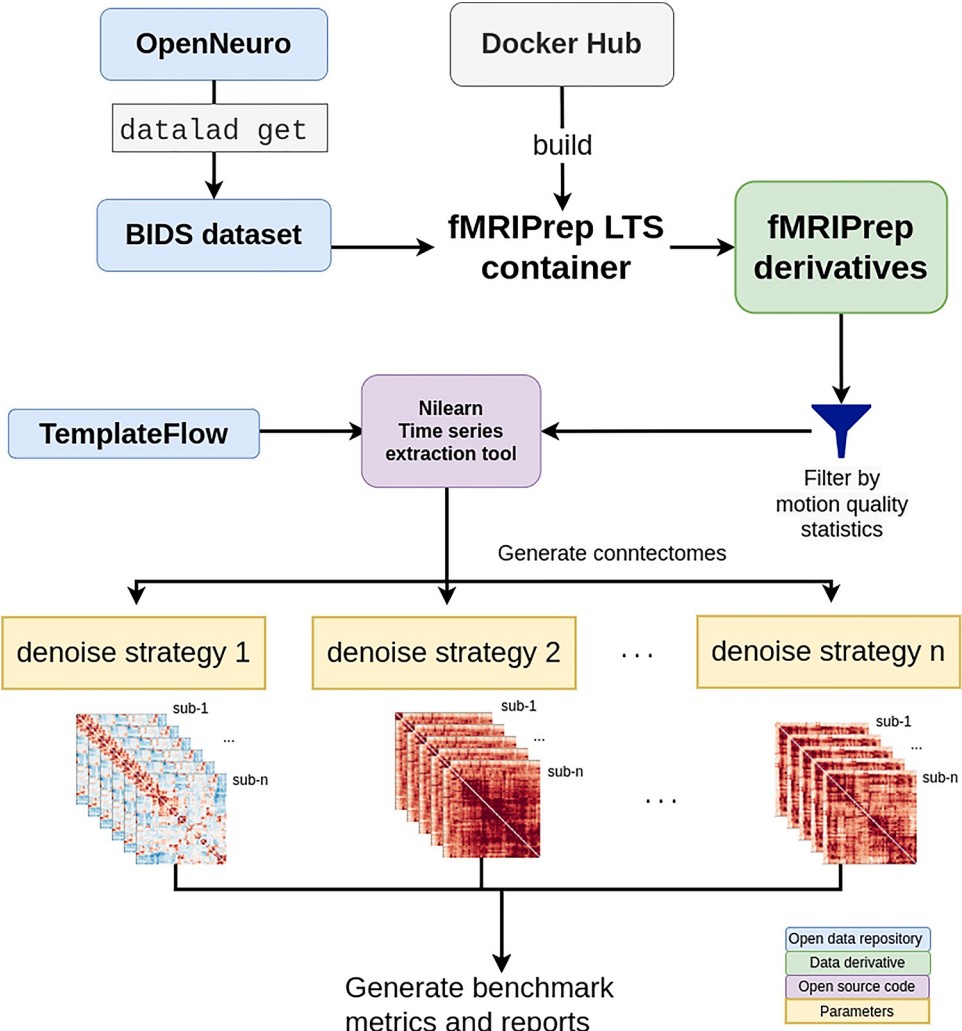

**Fig 2. Denoising benchmark workflow.** The denoising benchmark workflow expands on the workflow in Fig 1 (represented by the purple box). We retrieved the datasets from OpenNeuro through DataLad and all steps indicated with the arrows are implemented with bash scripts written for the SLURM scheduler. Atlases were either retrieved from the TemplateFlow archive or reformatted to fit the TemplateFlow format. The extracted time series, denoising metrics, and all metadata for generating the report are available on Zenodo [33].

in each dataset before and after the automatic motion quality control. Following this, we checked the difference in the mean framewise displacement of each sample and the sub-groups (Fig 3). In *ds000228*, there was still a significant difference (t(73) = -2.17, p = 0.033) in motion during the scan captured by mean framewise displacement between the child (M = 0.17, SD = 0.05, n = 51) and adult samples (M = 0.15, SD = 0.04, n = 24). In *ds000030*, the only patient group that showed a difference compared to control subjects (M = 0.12, SD = 0.04, n = 88) was the schizophrenia group (M = 0.16, SD = 0.05, n = 19; t(105) = -3.49, p = 0.033). There was no difference between the control and ADHD group (M = 0.12, SD = 0.05, n = 32; t(118) = 0.04, p = 0.966), or the bipolar group (M = 0.13, SD = 0.05, n = 29; t(115) = -1.24, p = 0.216). In summary, children moved more than adults, and subjects with schizophrenia moved more than controls.

**Table 3. Sample demographic information before and after removing subjects with high motion.**

| | | *ds000228* | | | *ds000030* | | | | |
|---|---|---|---|---|---|---|---|---|---|
| | | full sample | adult | child | full sample | control | ADHD | *bipolar* | schizophrenia |
| Before removal of high motion subjects | N (female) | 155 (84) | 33 (20) | 122 (64) | 212 (98) | 106 (54) | 35 (17) | 41 (19) | 30 (8) |
| | Mean Age (SD) | 10.6 (81) | 24.8 (5.3) | 6.7 (2.3) | 33.2 (9.3) | 31.8 (8.9) | 32.5 (10.2) | 34.7 (8.9) | 37.2 (9.2) |
| | Age Range | 3.5–39.0 | 18–39 | 3.5–12.3 | 21–50 | 21–50 | 21–50 | 21–50 | 22–49 |
| After removal of high motion subjects | N (female) | 75 (38) | 24 (14) | 51 (24) | 168 (79) | 88 (46) | 32 (14) | 29 (15) | 19 (4) |
| | Mean Age (SD) | 12.2 (8.4) | 23.6 (4.1) | 6.9 (2.4) | 31.7 (8.9) | 30.5 (8.2) | 32.3 (10.3) | 32.5 (8.3) | 35.2 (10.0) |
| | Age Range | 3.6–31.0 | 18–31 | 3.6–11.5 | 21–50 | 21–50 | 21–50 | 21–*48* | 22–49 |
| Number of subjects excluded (female) | | 80 (46) | 9 (6) | 71 (40) | 44 (19) | 18 (8) | 3 (3) | 12 (4) | 11 (4) |
| % of subjects excluded within groups | | 52% | 27% | 58% | 21% | 17% | 9% | 29% | 37% |

We also examined the differences between male and female in the control groups of the two datasets: the adult sample for *ds000228* and healthy control for *ds000030*. In *ds000228*, we found no significant differences (male: M = 0.16, SD = 0.04; female: M = 0.14, SD = 0.05; t(22) = 1.19, p = 0.249). In *ds000030* we found the male sample (M = 0.13, SD = 0.04) showed higher mean framewise displacement than the female sample (M = 0.11, SD = 0.04; t(86) = 2.17, p = 0.033).

Due to the imbalanced samples per group and low number of subjects in certain groups after the automatic motion quality control, we collapsed all groups within each dataset to avoid speculation on underpowered samples in the results. For a breakdown of each metric by atlas, please see the supplemental Jupyter Book [31].

**Most denoising strategies converged on a consistent average connectome structure.** With the benchmark workflow in place, we first aimed to assess the overall similarity between connectomes generated from each denoising strategy. We calculated Pearson's correlations between connectomes generated from all strategies presented in the benchmark (Fig 4). The connectome correlation pattern across denoising strategies was similar in both datasets. Overall, the strategies displayed at least moderate similarity with each other, with Pearson's correlations above 0.6. There were two large clusters of highly-related strategies, driven by the presence (or lack) of global signal regression. Within each cluster of strategies, the correlations amongst the strategies were strong, with values above 0.9. baseline and aroma did not fit well in either of the two clusters, indicating that denoising generally impacts the connectome structure, and that the ICA-AROMA might be sensitive to different sources of noise, compared to those captured by other strategies in the benchmark.

**Loss in temporal degrees of freedom varied markedly across strategies and datasets.** In previous research, the loss of temporal degrees of freedom has shown an impact on the subsequent data analysis. Higher loss in temporal degrees of freedom can spuriously increase functional connectivity [35]. Volume censoring-based and data-driven strategies (ICA-AROMA and some variations of CompCor) introduce variability to degrees of freedom and can bias group level comparisons [6].

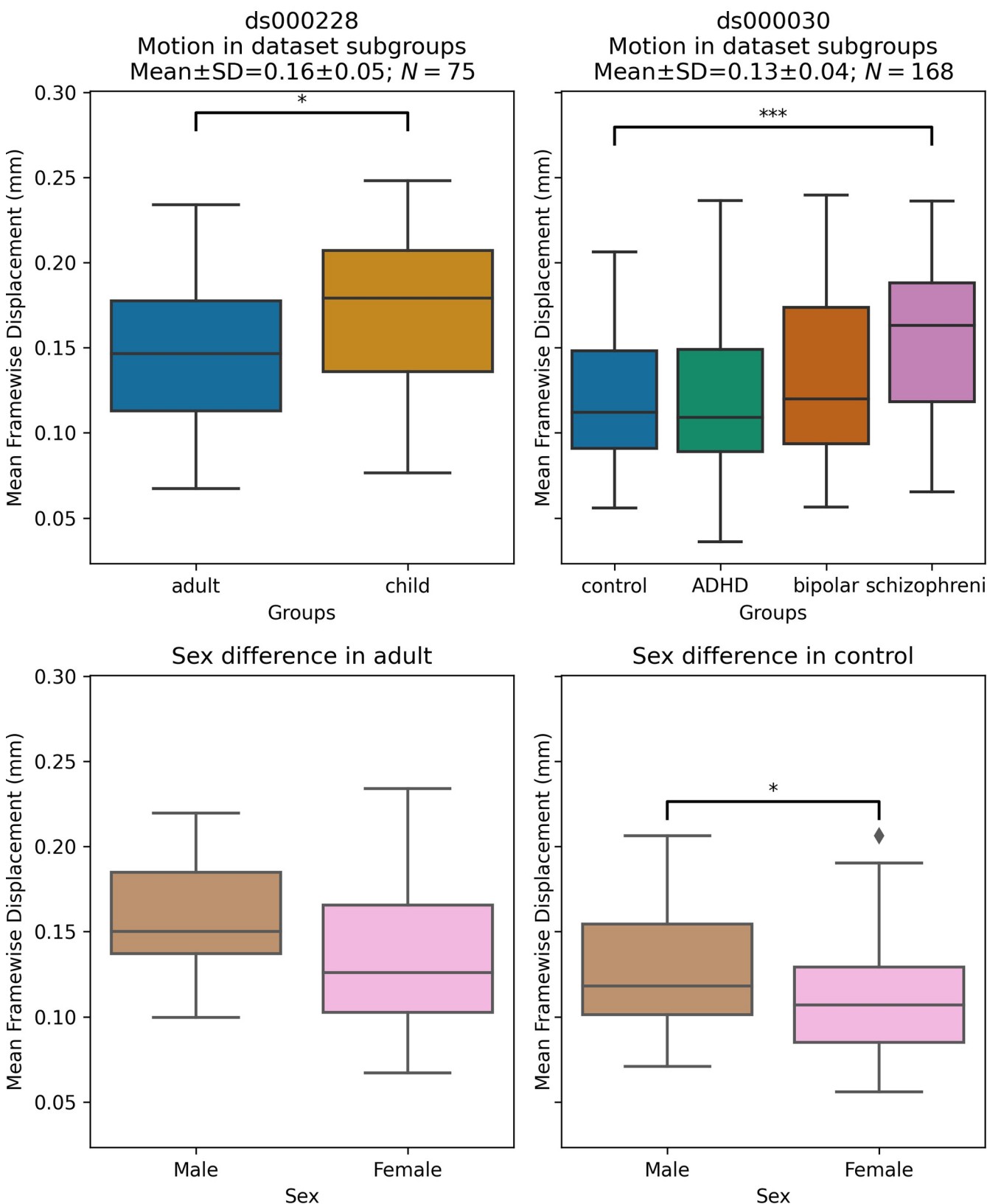

**Fig 3. Mean framewise displacement of each dataset.** To evaluate the metrics in a practical analytic scenario, we excluded subjects with high motion while preserving 1 minute of data for functional connectivity calculation: gross mean framewise displacement > 0.25 mm, above 80.0% of volumes removed while

scrubbing with a 0.2 mm threshold. In ds000228, the child group still had higher motion compared to the adult groups. In **ds000030**, where all subjects were adults, the control group only showed significant differences in motion with the schizophrenia group. In both datasets, the sample sizes from each group were highly imbalanced (see Table 3), hence no between group differences were assessed in quality metrics analysis.

The loss of temporal degrees of freedom is the sum of the number of regressors used and censored volume lost. Depending on the length of the scan, the number of discrete cosine-basis regressors can differ given the same repetition time (TR). The two datasets we analyzed contain different numbers of discrete cosine-basis regressors (*ds000228*: 4; *ds000030*: 3) due to difference in time series length (*ds000228*: 168; *ds000030*: 152). The `simple` and `simple+gsr` strategies include the same amount of head motion and tissue signal regressors between the two datasets (`simple`: 26, `simple+gsr`: 27). For volume censoring strategies, we observed a higher loss in volumes in *ds000228*, compared to *ds000030*. (number of excised volumes at 0.5 mm: *ds000030*: 2.5(4.4) range = [0 21], *ds000228*: 9.3(8.8) range = [0 30]; number of excised volumes at 0.2 mm: *ds000030*: 29.4(30.1) range = [0 110], *ds000228*: 53.0(34.1) range = [1 130]. `compcor` also showed variability in numbers of regressors when using all components that explain 50% of signal variance (number of CompCor regressors: *ds000030*: 47.9(3.9) range = [35 54], *ds000228*: 42.5(8.9) range = [5 58]). ICA-AROMA regressors in strategy `aroma` showed variability in numbers of regressors (number of ICA-AROMA regressors: *ds000030*: 16.0(4.6) range = [6 29], *ds000228*: 20.9(6.3); range = [7 38]). The average loss in temporal degrees of freedom is summarized in Fig 5.

The loss of degrees of freedom per strategy varied across the two datasets shown in the benchmark. The two datasets showed different loss of degrees of freedom in scrubbing-based strategies, while using the same gross motion-based exclusion criteria. This was expected, as the amount of motion between time points was higher in ds000228. The loss in degrees of freedom was most prominent in the group of children in this dataset (see S1 Fig). `compcor` did

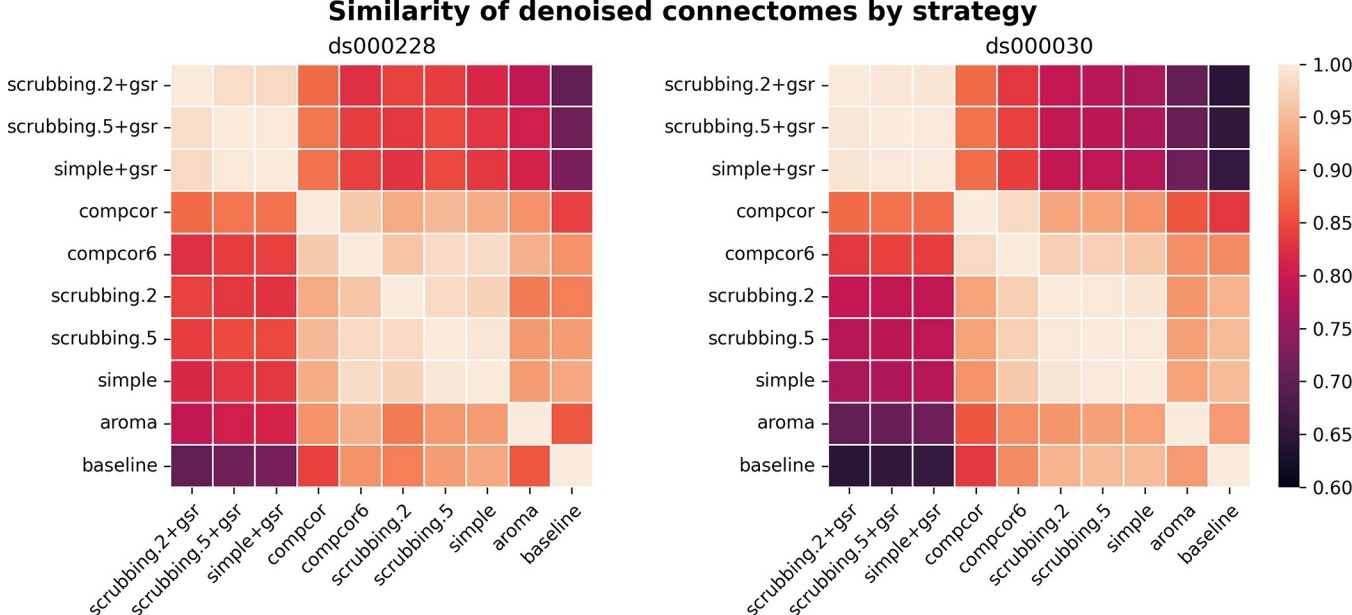

**Fig 4. Similarity of denoised connectomes.** For each parcellation scheme, we computed a correlation matrix across connectomes generated with the ten strategies. These correlation matrices were then averaged across the parcellation schemes within each dataset. Two large clusters of strategies emerged: with versus without global signal regression, with fairly high similarity in connectomes within each cluster.

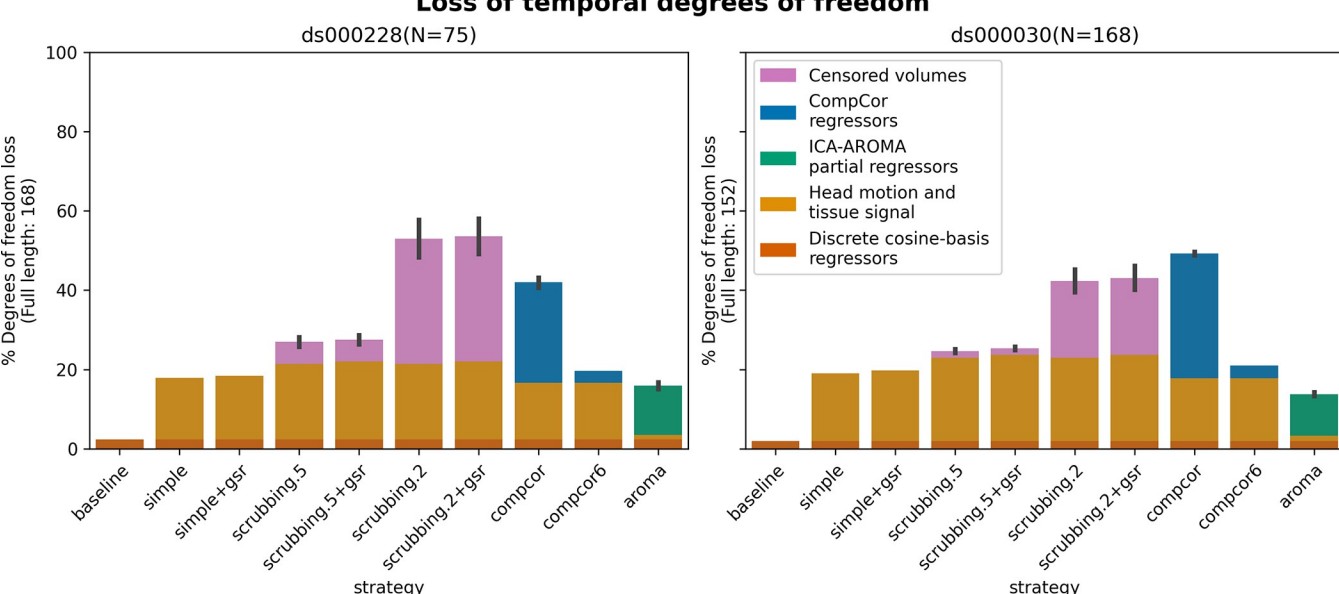

**Fig 5. Percentage of loss in temporal degrees of freedom according to strategy and dataset.** Bars show the average percentage of the number of regressors to the length of the scan amongst all subjects. Error bars indicate 95% confidence interval. The two datasets contain different numbers of discrete cosine-basis regressors (ds000228: 4; ds000030: 3). compcor (anatomical CompCor extracted from a WM/CSF combined map, cut off at 50% variance) and ICA-AROMA-based strategies (aroma) show variability depending on the number of noise components detected. The same figure with each dataset broken down by subgroup is in S1 Fig. The loss of degrees of freedom of the full dataset before filtered by movement is in S2 Fig.

not always have a lower loss of degrees of freedom in *ds000030*, and was actually higher in the bipolar subgroup (see S1 Fig). aroma had the lowest loss of temporal degrees of freedom in *ds000030*. Best practices for denoising will thus potentially differ depending on the characteristics of the subgroups included in a study, although the overall behavior of the different methods were consistent across datasets and subgroups.

**Quality control / functional connectivity (QC-FC) showed a heterogeneous impact of denoising strategies based on data profile.** The denoising methods should aim to reduce the impact of motion on the data. To quantify the remaining impact of motion in connectomes, we adopted a metric proposed by Power and colleagues [34] named quality control / functional connectivity (QC-FC). QC-FC is a partial correlation between mean framewise displacement and functional connectivity, with age and sex as covariates. Significance tests associated with the partial correlations were performed. P-values below the threshold of $\alpha = 0.05$ were deemed significant.

Scrubbing-based strategies consistently performed better than the baseline in both datasets. In *ds000228*, the most effective method according to QC-FC was scrubbing.5 (scrubbing at a liberal threshold), followed by scrubbing.2 and simple. All the GSR counterparts of the methods had slightly higher residual motion. Amongst all the data-driven methods, compcor performed the best. compcor6 and aroma performed close to baseline. In *ds000030*, the best performing method was compcor, followed by scrubbing.2 (aggressive scrubbing). The simple and scrubbing.5 methods performed similarly as very few volumes were censored with a liberal threshold, and the GSR variations (simple+gsr and scrubbing.5+gsr) performed better than baseline (see Fig 6). simple performed close to the baseline in terms of the number of edges correlated with motion. The aroma and compcor6 strategies were better than baseline. The average percentage of significant QC-FC

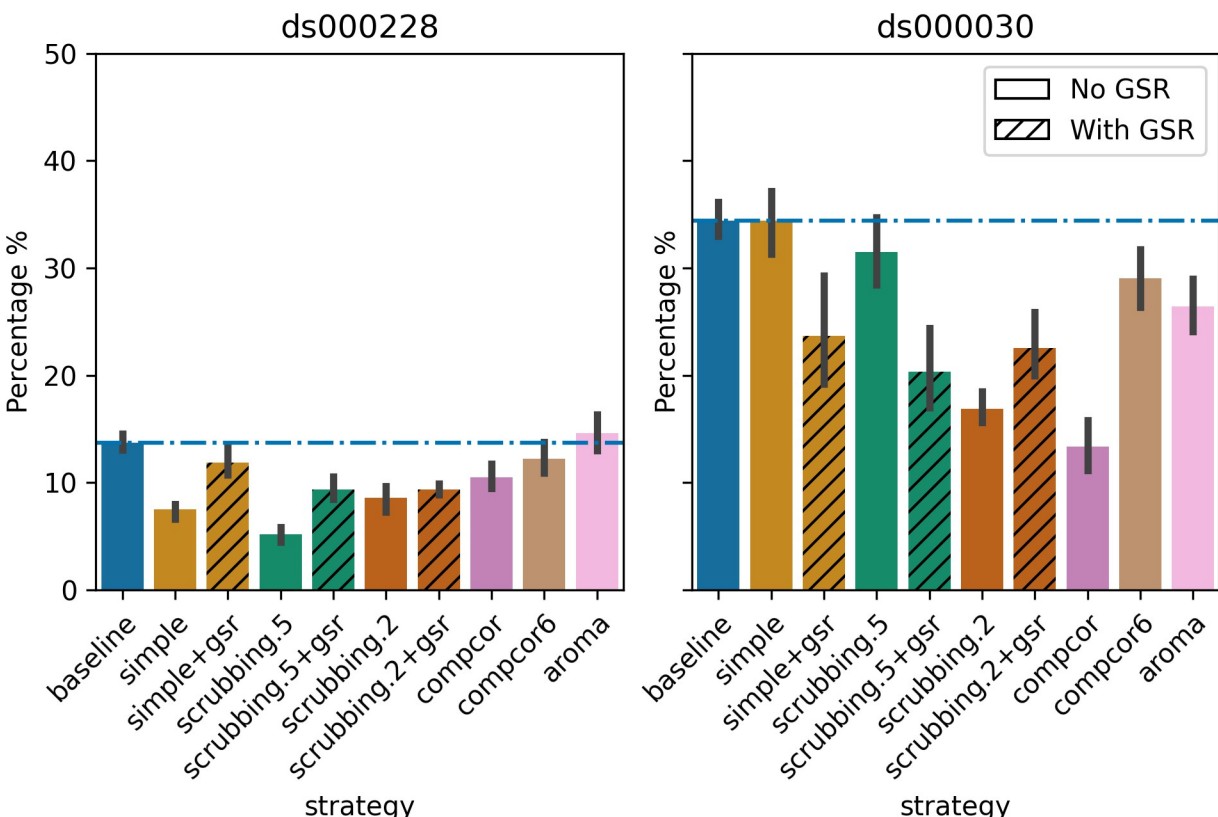

**Fig 6. Significant QC-FC in connectomes.** Average percentage of edges significantly correlated with mean framewise displacement are summarized across all atlases as bar plots. Error bars represent the 95% confidence intervals of the average. The horizontal line represents the baseline. A lower percentage indicates less residual effect of motion after denoising on connectome edges. Significant QC-FC associations were detected with p<0.05, uncorrected for multiple comparisons. A version of the figure using false-discovery-rate correction for multiple comparisons can be found in supplemental Jupyter Book.

and the average median of absolute value of QC-FC are presented in Figs 6 and 7. In summary, based on a QC-FC evaluation, diverse strategies performed quite differently based on the dataset used for evaluation.

**Scrubbing-based strategies decreased distance-dependent effects of motion.**   The impact of motion on functional connectivity has been reported to be higher for brain parcels closer to each other in space [25]. To determine the residual distance-dependent effects of subject motion on functional connectivity (DM-FC), we calculated a correlation between the Euclidean distance between the centers of mass of each pair of parcels [25] and the corresponding QC-FC correlations. We reported the absolute DM-FC correlation values and expected to see a general trend toward zero correlation after denoising.

All strategies performed better than the baseline in both datasets (Fig 8). We observed a trend consistent across both datasets, whereby strategies `scrubbing.2` and `scrubbing.2+gsr` were the most effective in reducing the correlation. `aroma` also performed consistently well in both datasets, ranked after `scrubbing.2`. In *ds000228*, `simple` was the least effective strategy for reducing distance dependency. Data-driven methods showed similar results to each other. `scrubbing.5` and `simple` greatly benefited from adding GSR in the

## Medians of absolute values of QC-FC

**Fig 7. Medians of absolute values of QC-FC.** Median of absolute value of QC-FC, averaged across all atlases of choice. Error bars represent the confidence intervals of the average at 95%. Low absolute median values indicate less residual effect of motion after denoising. The horizontal line represents the baseline. Results observed with absolute QC-FC values are consistent with the percentage of edges with significant QC-FC associations, as reported in Fig 6.

regressors. In *ds000030*, the difference between `scrubbing.2` and other strategies was bigger than in *ds000228*, with the remainder performed similarly with each other. The impact of GSR was small with the exception of `scrubbing.2+gsr`. In summary, we observed similar trends across strategies between the two datasets, yet with differences in the magnitude of correlations. All strategies reduced the correlation lower than the baseline. Consistent with the literature, scrubbing strategies were the best at reducing distance dependency.

**Global signal regression increases network modularity.** Confound regressors have the harmful potential to remove real signals of interest as well as motion-related noise. To evaluate this possibility, we examined the impact of denoising strategies on a common graph feature, network modularity, generally regarded as a key feature of biological network organization [4]. Network modularity was quantified using the Louvain method for graph community detection [36]. We computed the partial correlation between subjects' modularity values and mean framewise displacement, using age and sex as covariates, following the implementation of Power and colleagues [34].

The inclusion of global signal regressors increased average Louvain network modularity in both datasets (Fig 9, top panel). The remaining strategies performed as follows in both datasets, from best to worst: `compcor`, `scrubbing.2`, `scrubbing.5`, `simple`,

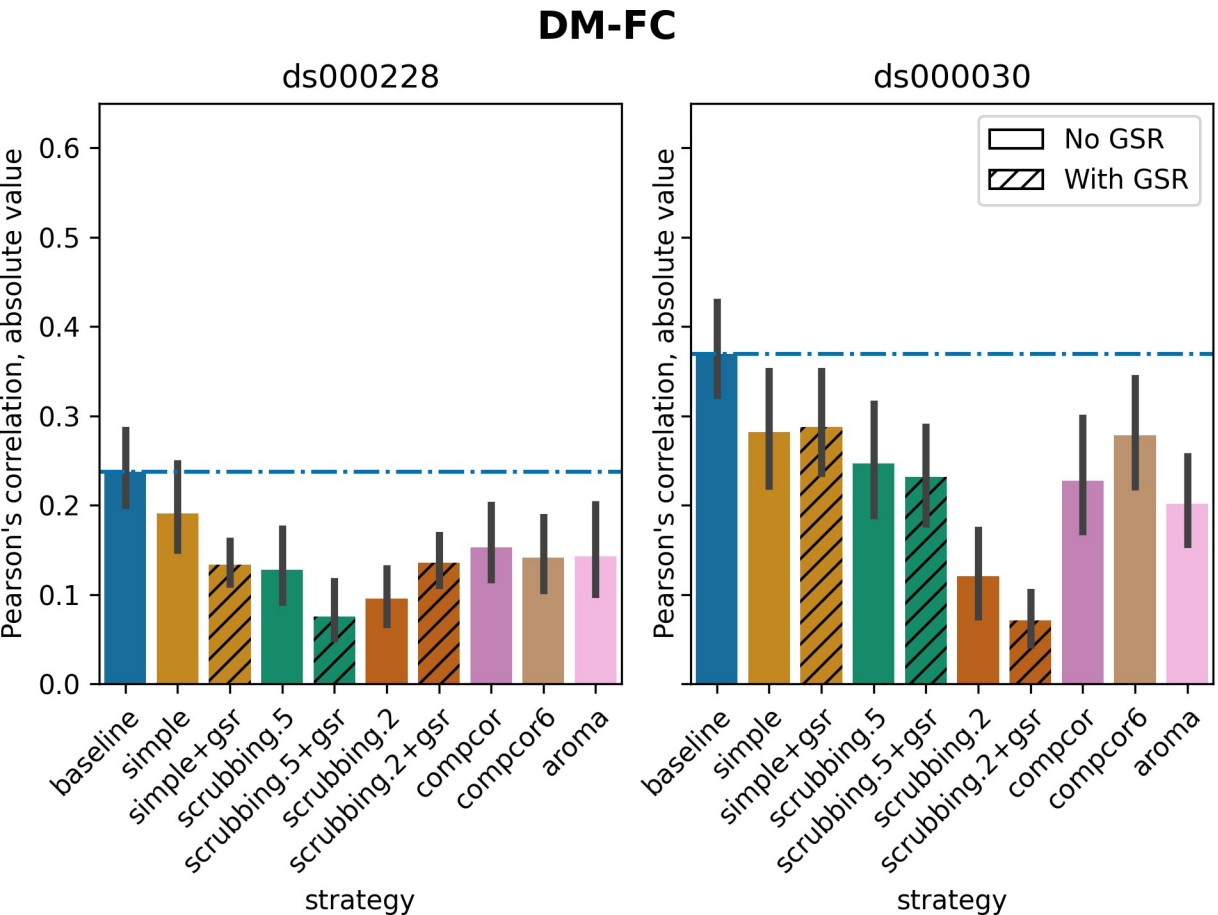

**Fig 8. Residual distance-dependent effects of subject motion on functional connectivity.** Average of absolute value of Pearson's correlation between the Euclidean distance between node pairs and QC-FC, indicating distance-dependent of motion after denoising. A value closer to zero indicates less residual effect of motion after denoising. Error bars represent the standard deviation. The horizontal line represents the baseline. Strategies scrubbing.2 and scrubbing.2+gsr were the most effective in reducing the correlation in both datasets.

compcor6, and aroma. In both datasets, aroma performed almost at the similar level as the baseline. We found fixed results in the ability of denoising in reducing the impact of motion on modularity (Fig 9 lower panels). In *ds000228*, we see simple and scrubbing.5 reducing the impact of motion. In *ds000030*, only scrubbing.2 performed better than baseline. In both datasets, the data-driven strategies and strategies with GSR performed consistently worse than baseline. The overall trend across strategies is similar to QC-FC with the exception of the baseline strategy (see Fig 6 and 7). The reason behind this observation could be a reduction of variance in the Louvain network modularity metric for GSR-based denoising strategies. We plotted the correlations of baseline, scrubbing.2, scrubbing.2+gsr from one parcellation scheme (DiFuMo 64 components) from *ds000030* to demonstrate this lack of variance (see Fig 10).

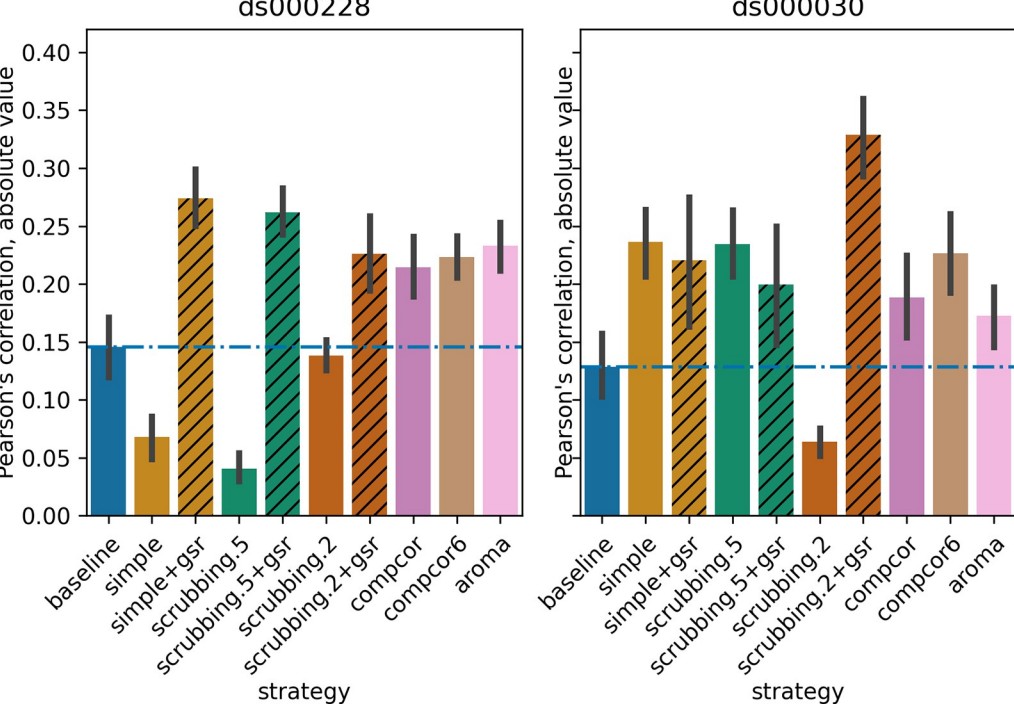

**Fig 9. Network modularity measures.** Top: Average Louvain network modularity of all connectomes after denoising. Error bars represent the standard deviation. The horizontal line represents the baseline. In both datasets, strategies including the global signal regressor(s) have higher modularity values. Bottom: Average Pearson's correlation between mean framewise displacement and Louvain network modularity after denoising. A value closer to zero indicates less residual effect of motion after denoising.

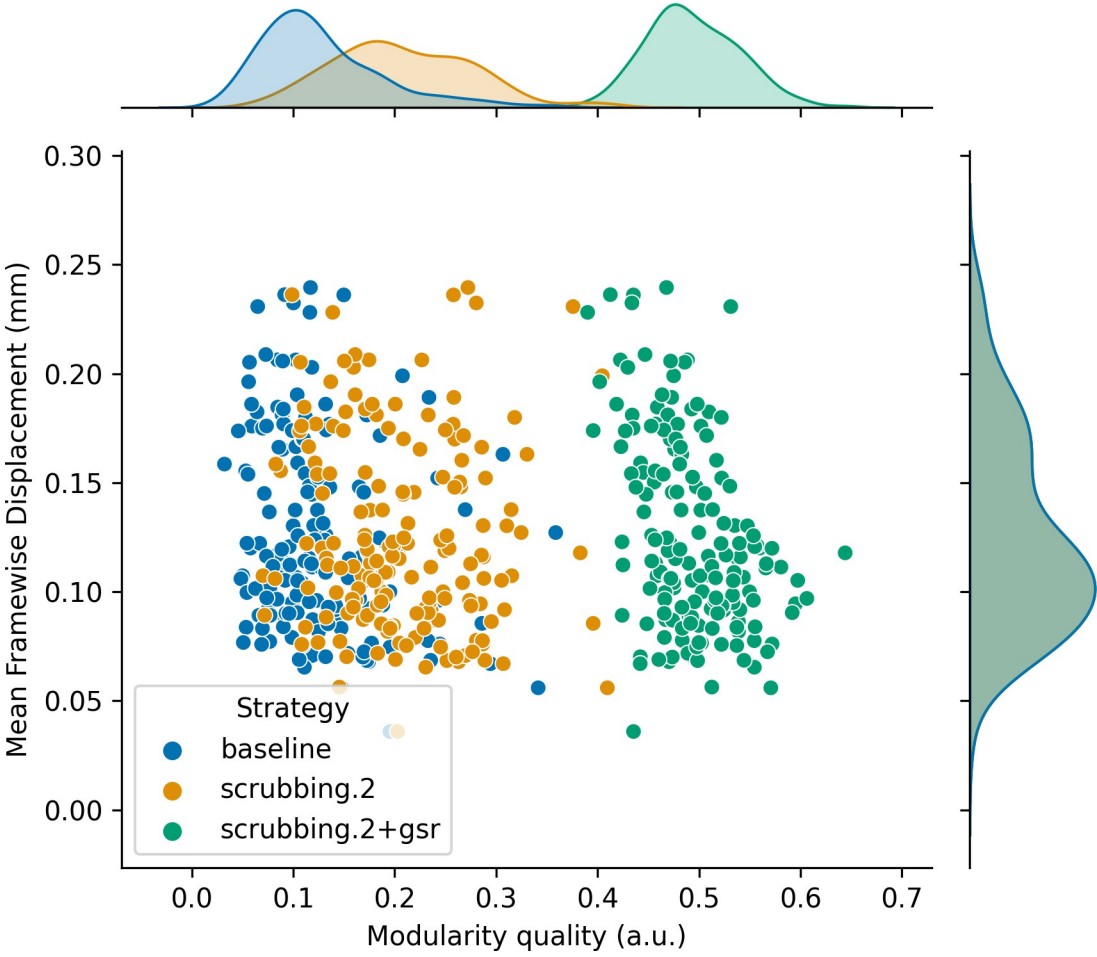

**Fig 10. Correlation between mean framewise displacement and Louvain network modularity after denoising.** We observed a lack of variance in Louvain network modularity, and shrinkage of the distribution with the inclusion of GSR. Due to the lack of variability, assessing residual motion in network modularity might not be a good metric to evaluate the quality of connectivity data.

### Data-driven denoising strategies showed inconsistent evaluation outcomes between two fMRIPrep versions

Different versions of the same software could produce differences in the outcomes of our denoising evaluation. To gain insight into the stability of fMRIPrep, we examined whether a few key observations from fMRIPrep 20.2.1 LTS remained salient in fMRIPrep 20.2.5 LTS, specifically:

1. High loss of temporal degrees of freedom for `scrubbing.2` in *ds000228* and `compcor` for *ds000030*.

2. `aroma` performed close to `baseline` in QC-FC for *ds000228*.

3. `simple` performed close to `baseline` in QC-FC for *ds000030*.

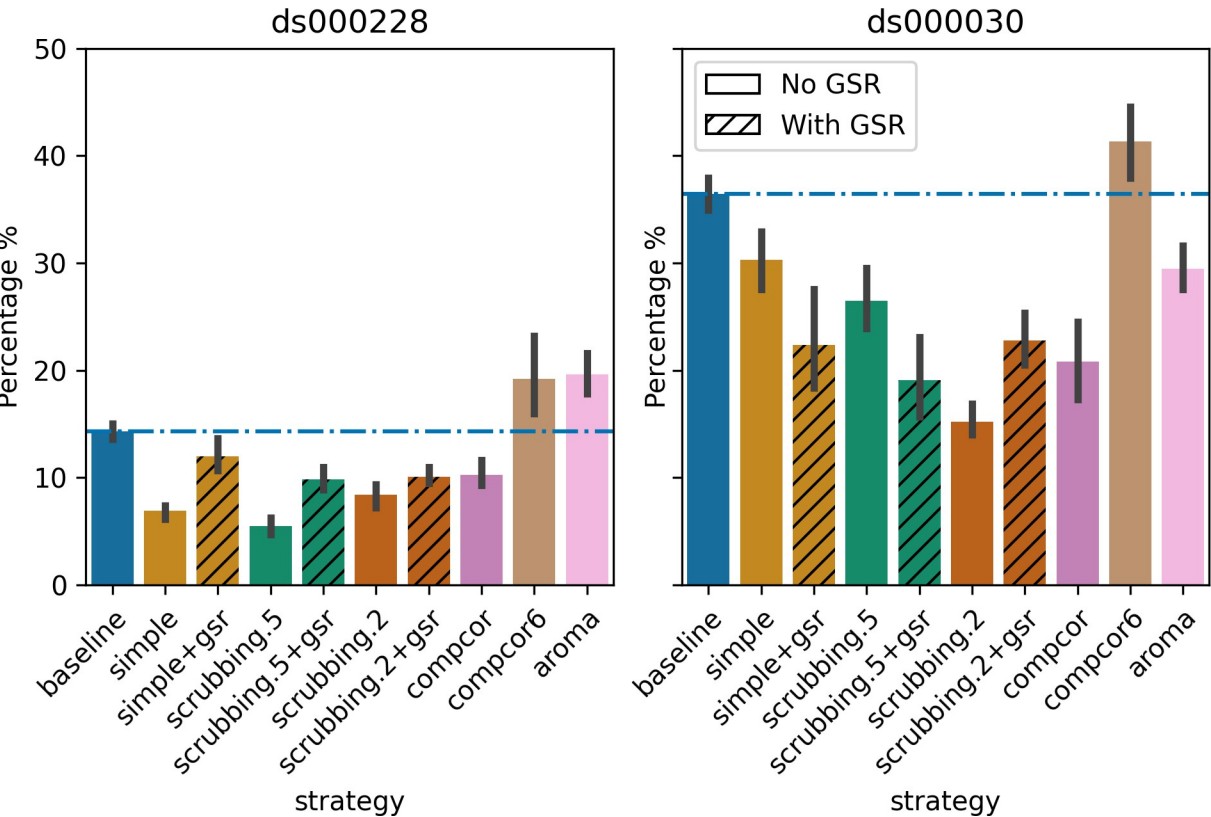

**Fig 11. Significant QC-FC in connectomes compiled from 20.2.5 LTS.** Average percentage of edges significantly correlated with mean framewise displacement are summarized across all atlases as bar plots. Error bars represent the 95% confidence intervals of the average. The horizontal line represents the baseline. Lower values indicate less residual effect of motion after denoising. Data-driven denoising strategies showed inconsistent patterns compared to the same metric generated from 20.2.1 LTS outputs (Fig 6).

4. `scrubbing.2` and `scrubbing.2+gsr` were the best strategies to reduce DM-FC.

5. GSR-enabled strategies showed higher network modularity.

Observations 1, 4, and 5 from 20.2.5 LTS were consistent with results from 20.2.1 LTS. The results of QC-FC demonstrated similar overall trends in 20.2.5 LTS, but with aroma performing worse than `baseline` for *ds000228* (observation 2) and `simple` performing better than baseline for *ds000030* (observation 3) (see Fig 11). Inconsistency in outcomes across the two fMRIPrep versions were found in strategies with data-driven noise components. In version 20.2.5 LTS, and unlike 20.2.1 LTS, `comcpor6` performed worse than the `baseline` in

**Table 4. Key observations compared between datasets and fMRIPrep versions.**

|  | ds000228 | | ds000030 | |
|---|---|---|---|---|
|  | **20.2.1** | **20.2.5** | **20.2.1** | **20.2.5** |
| Highest loss of temporal degrees of freedom | scrubbing.2 | scrubbing.2 | compcor | compcor |
| Worst performing in QC-FC | aroma | aroma and compcor6 | simple | compcor 6 |
| GSR-enabled strategies showed higher network modularity | yes | yes | yes | yes |

metric QC-FC for both datasets. In *ds000228*, `aroma` was the second worst performing strategy. For *ds000030*, the strategies with no data-driven noise components showed better performance in 20.2.5 LTS ([Fig 11]) than 20.2.1 LTS (see [Fig 6]). The key observations and the comparisons are summarized in [Table 4].

## Discussion

We aimed to create a re-executable benchmark to provide guidelines and accessible tools for denoising resting state functional connectivity data. The re-executable benchmark showed most denoising strategies, such as scrubbing-based strategies, `simple`, and strategies with GSR, performed in line with the literature. `aroma` showed an advantage in low degrees of freedom lost, while only performing relatively well in DM-FC amongst all quality metrics. The metrics performed consistently across the software versions with a marked exception in the data-driven denoising strategies (`compcor6, aroma`). This result demonstrates the necessity of distributing an executable research object for methods development and software testing, and providing accurate guidelines to users over time.

### The load_confounds and load_confounds_strategy APIs

The standardized APIs `load_confounds` and `load_confounds_strategy` are the core elements of the re-executable denoising benchmark. The APIs provide an easy way to implement classic denoising strategies from the literature, and can reduce the effort required, as well as errors, when using these strategies. Having clear and concise code also facilitates reuse and sharing of the denoising strategy used in a particular study, which improves reproducibility of science.

The new APIs developed for this project have been integrated in an established, popular software library, Nilearn [11]. The implementation of these APIs required other contributions to Nilearn and introduced new modules, in order to streamline the compatibility between the APIs and other data processing utilities. Specifically, we introduced a new module `nilearn.interfaces` dedicated to interacting with other neuroimaging software libraries and BIDS. We refactored the parameter `sample_mask` in all masker modules to allow volume censoring in the `signal.clean` function (move `sample_mask` to `transform` method in `maskers`, handle `sample_mask` in `signal.clean`: https://github.com/nilearn/nilearn/pull/2858). The masker modules implement a series of methods to convert 3D or 4D neuroimaging data into numerical arrays, for example extracting average time series from a brain parcellation. As a result, the outputs from `load_confounds` and `load_confounds_strategy`, as well as volume censoring information, can be directly ingested into all Nilearn masker objects. Thanks to these contributions, it is now possible to construct a complete Python-based fMRIPrep post-processing workflow with very concise code. Documentation for this workflow can be found in the Nilearn User Guide library [37], and users can adapt code from the Nilearn tutorial to implement denoising strategies with ease.

Similar functionality provided by the `load_confounds` and `load_confounds_strategy` APIs are included in other fMRIPrep-compatible fMRI processing software, such as C-PAC [13], XCP-D [38], and ENIGMA HALFpipe [39]. Unlike our APIs, which focus on retrieving denoising regressors only, these softwares provide denoising utilities bundled in a full preprocessing workflow. The denoising regressor retrieval steps amongst those softwares are therefore not reusable and more difficult to reproduce. Our APIs provide the advantage that users can easily reuse the denoising strategies. In fact, XCP-D has adopted our APIs in their code base. A limitation of our APIs is that the implemented denoising strategies are limited to those covered by the regressors included in fMRIPrep. With the constant development

## Ranking of all strategies per dataset per fMRIPrep version

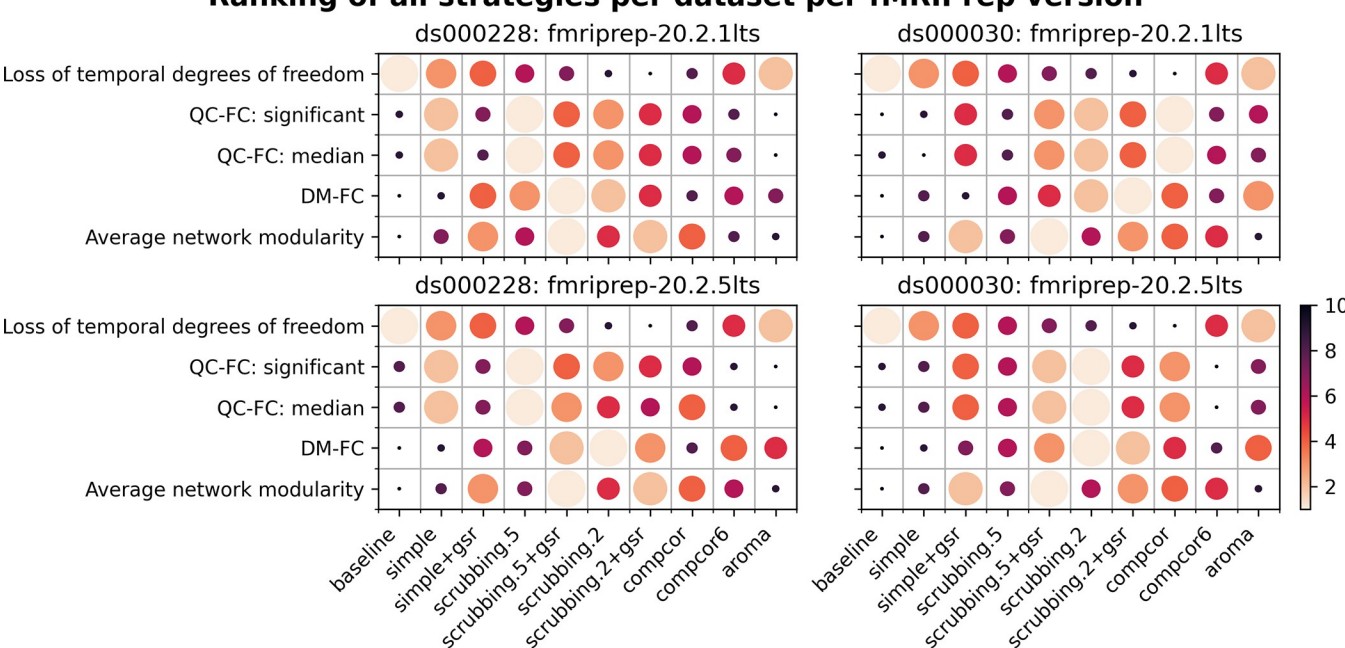

**Fig 12. Ranking of all denoising strategies across multiple performance metrics.** We ranked strategies across four metrics from best to worst. Larger circles with brighter color represent higher ranking. Metric "correlation between network modularity and motion" has been excluded from the summary as it is potentially a poor measure. Loss of temporal degrees of freedom is a crucial measure that should be taken into account alongside the metric rankings. A clear trade-off is apparent between loss in degrees of freedom and the quality of denoising, so no overall ranking of methods is derived from this analysis—see text for a summary of key takeaways.

of denoising strategies, what the APIs provide will always lag behind the advancement of the field. However, as a trade-off, we can ensure the quality and robustness of the implementation.

### Denoising strategy

In order to summarize our results, we created a table ranking strategies from best to worst, based on four benchmark metrics, across datasets and fMRIPrep versions (see Fig 12).

The ranking of the loss of temporal degrees of freedom is an important consideration accompanying the remaining metrics, as any denoising strategy aims at a particular trade-off between the amount of noise removed and the preservation of degrees of freedom for signals. Aside from the loss of temporal degrees of freedom, the `baseline` strategy consistently performs the worst, as expected, with the notable exception of `aroma` performing worst on QC-FC.

The `simple+gsr` strategy is not the best for any particular individual evaluation metric, but it performed consistently well across metrics, datasets and software versions. The loss in degrees of freedom `simple` (26 + number of cosine terms) and `simple+gsr` (27+number of cosine terms) used slightly more regressors than aroma, and had markedly lesser loss than `scrubbing` methods. `simple+gsr` is consistently better than other data-driven strategies, which makes it the best choice for analysis that requires low loss of degrees of freedom and also preserve continuous sampling time series (which is broken by `scrubbing`).

Scrubbing based strategies are the best when it comes to minimizing the impact of motion, with a cost of higher loss in degrees of freedom. We found that scrubbing with an aggressive 0.2 mm threshold (`scrubbing.2`) mitigates distance dependency well consistently,

regardless of the group of subjects. Despite excluding data with the same standard on both datasets, the child-dominant sample (*ds000228*) showed more volumes censored with the scrubbing strategy, and a liberal framewise displacement threshold showed sufficient ability to reduce the distance dependency of motion as observed in the original study of the strategy [25]. In a sample with higher motion, such as *ds000228*, a liberal scrubbing threshold reduced the impact of motion and performed similarly with a higher threshold. Taking the loss of degrees of freedom into consideration, we recommend a liberal scrubbing threshold rather than scrubbing with a stringent threshold for datasets with marked motion.

For the two anatomical CompCor strategies, `compcor` performs better than `compcor6`. The performance of `compcor6` is also not consistent across software versions in both datasets. However, `compcor` introduces large variability into the loss of degrees of freedom. In *ds000228*, the loss in temporal degrees of freedom is even higher than scrubbing with a stringent threshold. This result is consistent with the observation of Parkes and colleagues [7] that anatomical CompCor is not sufficient for high motion data. Moreover, this observation puts one of the rationales in the original study, i.e., to reduce the loss in degrees of freedom, in question [26]. In the absence of physiological recordings, our benchmark is not suitable to examine another property of CompCor, that is the ability to remove unwanted physiology signals [26]. The datasets do not include physiology measures to perform alternative strategies such as RETROICOR to mitigate physiology signals explicitly.

In our results, `aroma` shows similar performance with the `simple` strategy across most metrics, with the exception of DM-FC (where it performs well). This strategy also featured a very low loss of degrees of freedom, making it a "gentle" denoising approach. Previous literature has recommended adding GSR as part of the ICA-AROMA regressors [6,7]. An early version of this work did include an ICA-AROMA+GSR strategy, which performed very poorly (see Fig 4–9, 11, and 12 from https://www.biorxiv.org/content/10.1101/2023.04.18.537240v1. full). This is a known consequence of implementation choices made in fMRIprep, which departs from the original recommended implementation of ICA-AROMA+GSR [27; seeS1 Text Annex D for detailed explanation]. We strongly recommend fMRIPrep users to avoid fMRIPrep-generated GSR when using the ICA-AROMA strategy. It is also worth noting that fMRIPrep will drop the support for ICA-AROMA from version 23.1.0 (https://github.com/ nipreps/fmriprep/issues/2936).

Strategies including GSR produced connectomes with higher network modularity compared to their counterparts without GSR. There is no systematic trend of whether GSR improves the denoising strategies based on the remaining impact of motion. The result is consistent with the fact that global signal regression increases the number of negative connections in a functional connectome (see Nilearn examples visualizing connectomes with and without global signal regression [37]) by shifting the distribution of correlation coefficients to be approximately zero-centered [40]. A clear benefit of GSR is thus to emphasize the network structure, but its benefits for denoising can vary. Some strategies, such as `simple`, seem to benefit greatly from the addition of GSR.

Finally, we would like to address a few limitations of the evaluation on denoising strategies.

1. **Loss of statistical power for downstream analysis with stringent motion-based exclusion criteria**: The current evaluation was performed on datasets after excluding subjects with gross in-scanner motion, as per existing literature [6,7]. The aim of the exclusion allowed evaluation of denoising strategies on the mitigation of artifacts due to micro-movements. However, the exclusion criteria would result in loss of power in downstream analysis in certain demographics, and was particularly apparent here for the "child" group in ds0000228 and the "schizophrenia" group in ds000030. We encourage researchers to take this potential

loss in sample size into account for selecting an appropriate denoising strategy for their study. Our Neurolibre companion offers comparisons of mean framewise displacement and loss of temporal degrees of freedom at two different motion exclusion thresholds or without exclusions, for reference.

2. **Unclear implication for datasets with lower TR or multiband scanning sequence**: The current benchmark used two datasets with TR of 2 to 2.5 seconds, thus the conclusions are limited to fMRI datasets with a similar scanning sequence. For multi-band fMRI sequences with shorter TR, there may be different motion concerns such as respiratory motion [41,42]. It would be of the community's interest to explore the current workflow on multi-band fMRI datasets, and including physiology related denoising strategies and different quality control metrics [43] to address the current limitations.

3. **Heterogeneity of the datasets included in the benchmark**: The current observations were drawn from two datasets with 6 different subgroups. This is not comprehensive of all the possible demographic groups. We did not make the workflow a packaged software that can be executed on any generic data sets and the constraints will be discussed below. Instead, we provide instructions documenting the workflow in our Neurolibre companion material for interested researchers to implement on their own data.

### Re-executable research object

We created a re-executable denoising benchmark with two main outcomes. Firstly, we created a reusable code base that will ensure the robustness of the procedure. The current benchmark includes several parameters, from the choices of atlases, denoising strategies, fMRIPrep versions, to datasets. The code for connectome generation and denoising metric calculation is written as an installable Python library (https://github.com/SIMEXP/fmriprep-denoise-benchmark). Customized scripts to deploy the process for each combination of the parameters are also generated by reusable Python functions. The full workflow can be executed on the two benchmark datasets preprocessed by any release from the fMRIPrep LTS series. Secondly, we created an interactive Jupyter Book [19] hosted on NeuroLibre [20] for users to freely browse the results with finer details. All figures in this report can be rebuilt with the provided Makefile, handling data download and the report generation. Taken together, it is possible to reproduce the results of this manuscript, starting from raw data down to final figures, and update the entire manuscript on future releases of fMRIPrep, turning this research object into a living publication rather than a snapshot of current software destined for quick deprecation.

The current workflow is presented as a research object rather than a software due to the lack of generalizability on other datasets. For the analysis after fMRIPrep, there are two practical reasons for this choice. Firstly, compared to a piece of well packaged software, research objects allow more flexibility for changes for development. Secondly and most importantly, creating a clean, generalizable solution will require the data to be standardized. Although fMRIPrep outputs are standardized, the demographic information is coded differently across datasets. Currently the BIDS specifications do not impose restrictions on the label for phenotypic data, thus we had to manually harmonize the label for age, gender, and group information. As an alternative, full documentation to re-execute the workflow, from fetching datasets to running the analysis, is available as part of the research object.

There are additional benefits to creating a re-executable denoising benchmark. Although the code is not readily designed to process new datasets, it contains good prototypes for what could become different BIDS-apps for post processing [18]: a connectome generation BIDS-app and a denoising metric generation BIDS-app. BIDS-app is easier for user adoption under

the BIDS convention and can expand the scope of the benchmark from the two datasets shown here to any BIDS-compliant dataset. The process of creating this benchmark also provides valuable first hand information about runtime, and the impact of atlas choice on computational costs, which we did not cover here but has big practical implications. High dimensional probabilistic atlases require four times more RAM than discrete segment atlases. For metric generation, high dimensional atlases can have a runtime up to 24 hours compared to 1 hour for low dimensional atlases. There is thus a very concrete "reproducibility" cost which comes with high-resolution and probabilistic atlases. The issue is rarely mentioned regarding the reproducibility of science, yet can be a real obstacle to actual reproduction. Future editions of the workflow will be built with runtime optimization in mind and potentially improve the code base for upstream projects, such as fMRIPrep.

## Continual evaluation of software versions

Our benchmark results on two versions of the long-term support (LTS) release of fMRIPrep reveals similar trends in the metrics, but some inconsistency. Between the two datasets, *ds000228* showed more consistent results than *ds000030* across two LTS releases (see Fig 12). The marked difference in *ds000030* was likely the result of a bug fix implemented in 20.2.2LTS (See: https://github.com/nipreps/fmriprep/issues/2307, and #2444 in change log https://fmriprep.org/en/stable/changes.html#july-16-2021) and that *ds000030* had been reported as an affected dataset. The results from the data-driven strategies in both datasets demonstrated inconsistent relative difference when comparing to the baseline strategy. This piece of work is a new addition to the existing literature on the heterogeneity of results found through research software testing [12,14]. Beyond mere numerical instabilities, we show that the qualitative conclusions of an evaluation benchmark do not necessarily generalize to different software packages or even versions of the same package. Our results thus highlight the importance of continuous evaluation of research software at each major step of its life cycle.

Rebuilding this paper on future fMRIPrep releases can be used to perform such continuous evaluation for future releases of fMRIprep. This benchmark is thus a hybrid contribution, being as much research paper as it is a software development tool. We still recommend several aspects of improvements to better achieve this goal for future similar efforts. Firstly the API will need to be kept up to date with fMRIPrep releases. The current code will be applicable for 20.2.x series up to September 2024. For fMRIPrep release beyond the LTS version, as long as the API in Nilearn is maintained, the code used to generate all current reports can be applied to the same two datasets. With the high number of tunable parameters (denoise strategies, atlases, software versions), a framework allowing parameter configuration, such as Hydra (https://github.com/facebookresearch/hydra), would help better manage and expand the benchmark. The current benchmark generates jobs through metadata stored in python dictionaries. By adapting a framework like Hydra, one can deploy the benchmark analysis with a simplified interface.

Finally, we note that all of the components necessary to implement our reproducible benchmarks are generic, i.e. software containers, data versioning, open source code with standard APIs, Jupyter Books and the Neurolibre preprint server. Beyond the particularities of fMRI denoising and the fMRIprep implementation, we thus believe that this work proposes an approach to implement reproducible benchmarks that is widely applicable, and would likely be beneficial in all scientific fields with heavy reliance on computational tools.

## Conclusions

This work showcases the benefit of systematic evaluation on the impact of denoising strategies on resting state functional connectivity across datasets, and versions of the fMRIPrep preprocessing pipeline. With a standardized function to specify denoising strategy, we implemented a fully reproducible benchmark of denoising strategies for two datasets with varied characteristics, including age, motion level and the presence of clinical diagnoses. We would like to provide two strategy recommendations based on this benchmark, depending on a key consideration: whether preserving continuous sampling time series is needed (e.g. to train auto-regressive models) or not (e.g. to generate correlation coefficients across brain parcels). To preserve the continuous sampling property of time series, `simple+gsr` is the recommended strategy, especially for datasets with low motion, and appears to be robust across software versions. If continuous temporal sampling is not a priority, `scrubbing.5` is recommended for datasets with marked motion where denoising quality can be favored over loss of temporal degrees of freedom. The performance of aroma departed from the conclusions of previous denoising benchmark works and only performed well in one metric. The denoising benchmark also demonstrated differences in the performance of specific denoising strategies across multiple fMRIPrep versions. We hope that our benchmark provides useful insights on denoising strategies for the community and demonstrates the importance of continuous evaluation of denoising methods. Our benchmark also works as a proof of concept for re-executable quality assessments and a foundation for potential software for time series extraction and denoising strategy evaluation when a community solution for harmonizing demographic information emerges. Some elements and concepts of this project, such as Neurolibre, Jupyter Book and workflows are broadly applicable research computing practices and may be beneficial to implement reproducible benchmarks across different tools and research fields.

## Materials and methods

### Datasets

Dataset *ds000228* (N = 155) contains fMRI scans of participants watching a silent version of a Pixar animated movie "Partly Cloudy". The dataset includes 33 adult subjects (Age Mean(s.d.) = 24.8(5.3), range = 18–39; 20 female) and 122 child subjects (Age Mean(s.d.) = 6.7(2.3), range = 3.5–12.3; 64 female). T1w images were collected with the following parameters: TR = 2530 ms, TE = 1.64 ms, Flip Angle = 7˚, 1 mm isotropic voxels. BOLD images were collected with the following parameters: TR = 2000 ms, TE = 30 ms, Flip Angle = 90˚, 3 x 3 x 3.3 mm voxels. All images were acquired on a 3T Siemens Trio Tim Scanner. For more information on the dataset please refer to [16].

Dataset *ds000030* includes multiple tasks collected from subjects with a variety of neuropsychiatric diagnosis, including ADHD, bipolar disorder, schizophrenia, and healthy controls. The current analysis focused on the resting-state scans only. Scans with an instrumental artifact (flagged under column ghost_NoGhost in participants.tsv) were excluded from the analysis pipeline. Of 272 subjects, 212 entered the preprocessing stage. Demographic information per condition can be found in *Table 3* in the main text. T1w images were collected with the following parameters: TR = 2530 ms, TE = 3.31 ms, Flip Angle = 7˚, 1 mm isotropic voxels. BOLD images were collected with the following parameters: TR = 2000 ms, TE = 30 ms, Flip Angle = 90˚, 3 x 3 x 4 mm voxels. All images were acquired on a 3T Siemens Trio Tim Scanner.

### fMRI data preprocessing

We preprocessed fMRI data using fMRIPrep 20.2.1LTS and 20.2.5LTS through fMRIPrep-slurm (https://github.com/SIMEXP/fmriprep-slurm) with the following options:

```
-use-aroma \
-omp-nthreads 1 \
-nprocs 1 \
-random-seed 0 \
-output-spaces MNI152NLin2009cAsym MNI152NLin6Asym
-output-layout bids \
-notrack \
-skip_bids_validation \
-write-graph
-resource-monitor
```

For the full description generated by fMRIPrep, please see Neurolibre preprint [31]. We reported the primary outcomes using outputs from fMRIPrep 20.2.1LTS, and then investigated if the same conclusions can be observed in 20.2.5LTS.

### Choice of atlases

We extracted time series with regions of interest (ROI) defined by the following atlases: Gordon atlas [44], Schaefer 7 network atlas [45], Multiresolution Intrinsic Segmentation Template (MIST) [46] and Dictionary of Functional Modes (DiFuMo) [47]. All atlases were resampled to the resolution of the preprocessed functional data.

Since DiFuMo and MIST atlases can include networks with disjointed regions under the same label, we carried out further ROI extraction. Labels are presented with the original number of parcels. and we denote the number of extracted ROI in brackets. Gordon and Schaefer atlas parcels use isolated ROI, hence no further extraction was done. The Schaefer 1000 parcels atlas was excluded; regions were small enough that not all could be consistently resolved after resampling the atlas to the shape of the processed fMRI data.

- Gordon atlas: 333

- Schaefer atlas: 100, 200, 300, 400, 500, 600, 800

- MIST: 7, 12, 20, 36, 64, 122, 197, 325, 444, "ROI" (210 parcels, 122 split by the midline)

- DiFuMo atlas: 64 (114), 128 (200), 256 (372), 512 (637), 1024 (1158)

Processes involved here are implemented through Nilearn [11]. Time series were extracted using `nilearn.maskers.NiftiLabelsMasker` and `nilearn.maskers.NiftiMapsMasker`. Connectomes were calculated using Pearson's Correlation, implemented through `nilearn.connectome.ConnectivityMeasure`.

### Participant exclusion based on motion

We performed data quality control to exclude subjects with excessive motion leading to unusable data. In the current report, we use framewise displacement as the metric to quantify motion. Framewise displacement indexes the movement of the head from one volume to the next. The movement includes the transitions on the three axes ($x$, $y$, $z$) and the respective rotation ($\alpha$, $\beta$, $\gamma$). Rotational displacements are calculated as the displacement on the surface of a sphere of radius 50 mm [25]. fMRIPrep generates the framewise displacement based on the formula proposed in [25]. The framewise displacement, denoted as $FD_t$, at each time point $t$ is

expressed as:

$$FD_t = |\Delta d_{xt}| + |\Delta d_{yt}| + |\Delta d_{zt}| + |\Delta d_{\alpha t}| + |\Delta d_{\beta t}| + |\Delta d_{\gamma t}|.$$

To ensure the analysis is performed in a realistic scenario we exclude subjects with high motion [7] while retaining at least 1 minute of scan for functional connectome construction, defined by the following exclusion criteria: mean framewise displacement > 0.25 mm, above 80.0% of volumes removed while scrubbing with a 0.2 mm threshold.

## Confound regression strategies

Confound variables were retrieved using (i) a basic API that retrieves different classes of confound regressors, `nilearn.interfaces.fmriprep.load_confounds` (simplified as `load_confounds`); and (ii) a higher level wrapper to implement common strategies from the denoising literature, `nilearn.interfaces.fmriprep.load_confounds_strategy` (simplified as `load_confounds_strategy`). For documentation of the actual function, please see the latest version of Nilearn documentation (https://nilearn.github.io/stable/). The connectome generated from high-pass filtered time series served as a baseline comparison. The detailed 10 strategies and a full breakdown of parameters used in these strategies is presented in Table 3.

We evaluated common confound regression strategies that are possible through fMRIPrep-generated confound regressors and accessible through `load_confounds_strategy`. However, not all possible strategies from the literature are included. For example, ICA-AROMA + global signal regressor was not included, as the implementation in fMRIPrep is not in line with the original implementation(See ICA-AROMA related warning in https://fmriprep.org/en/20.2.1/outputs.html#confounds). Another excluded approach was commonly used by CONN combining scrubbing and aCompCor [48] because we want to focus on strategies corresponding to load_confounds_strategy and past benchmark literature. It can be implemented with load_confounds:

```
from nilearn.interfaces.fmriprep import load_confounds
confounds_simple, sample_mask = load_confounds(
  fmri_filenames,
  strategy = ["high_pass", "motion", "compcor", "scrub"],
  motion = "derivatives", scrub = 0, fd_threshold = 0.5,
  std_dvars_threshold = None,
  compcor = "anat_separated", n_compcor = 5)
```

## Signal denoising through linear regression

The filtered confounds and the corresponding preprocessed NIFTI images were then passed to the Nilearn masker generated with the atlas where the underlying function `nilearn.signals.clean` applied the regressors for denoising (see https://nilearn.github.io/stable/modules/generated/nilearn.signal.clean.html). S1 Text Annex E contains the mathematical operation implemented by the denoising procedure. The time series are then passed to `nilearn.connectome.ConnectivityMeasure` for generating connectomes

For scrubbing based strategies, the `nilearn.signals.clean` function censors the high motion time points before denoising with linear regression, known as the censoring approach. We did not use another common approach which is entering the high motion time points as one-hot encoders in the same linear regression with other confound regressors, known as the regression approach. The regression approach is equivalent to imputing the high motion time points with the average of the remaining time series. The benchmark assessed

fMRI functional connectivity metrics. The two approaches will produce numerically equivalent results. It's important to note that scrubbing strategy performed with either approach is a form of interruption of continuous time series, and will disrupt many operations such as, e.g., calculating a power spectrum.

## Evaluation of the outcome of denoising strategies

We first performed Pearson's correlations to understand the overall numerical similarities of the denoised connectomes across different strategies. For each parcellation scheme, we computed a correlation matrix across the thirteen strategies. These correlation matrices were then averaged across the parcellation schemes within each dataset. The averaged correlation matrices were reordered into blocks of clusters with the function `scipy.cluster.hierarchy.linkage`. The aim was to provide an overview of the similarity of connectomes generated with the strategies.

We then used selected metrics described in the previous literature to evaluate the denoising results [6,7]. After investigating the metrics with fMRIPrep version 20.2.1 LTS, we assessed whether the conclusions were consistent in 20.2.5 LTS.

**Loss in temporal degrees of freedom.**   The common analysis and denoising methods are based on linear regression. Using more nuisance regressors can capture additional sources of noise-related variance in the data and thus improve denoising. However, this comes at the expense of a loss of temporal degrees of freedom for statistical inference in further analysis. This may be an important point to consider alongside the denoising performance for researchers who wish to perform general linear model based analysis. Higher loss in temporal degrees of freedom can spuriously increase functional connectivity [35]. Volume censoring-based and data-driven strategies (ICA-AROMA and some variations of CompCor) introduce variability to degrees of freedom and can bias group level comparisons [6]. We calculate the number of regressors used and number of censored volume loss. Depending on the length of the scan, the number of discrete cosine-basis regressors can differ. The number of discrete cosine-basis regressors will be denoted as $c$ in the report ($c_{ds000228} = 4$, $c_{ds000030} = 3$). `Simple`, `simple+gsr`, `compcor6` are the strategies with a fixed number of degrees of freedom loss. `Scrubbing`, `compcor`, `aroma`, and `aroma+gsr` strategies show variability depending on the number of noise components detected.

**Quality control / functional connectivity (QC-FC).**   QC-FC [34] quantifies the correlation between mean framewise displacement and functional connectivity. This is calculated by a partial correlation between mean framewise displacement and connectivity, with age and sex as covariates. The denoising methods should aim to reduce the QC-FC value. Significance tests associated with the partial correlations were performed, and correlations with P-values below the threshold of $\alpha = 0.05$ deemed significant. A version of this analysis corrected for multiple comparisons using the false discovery rate [49] is available in the Neurolibre preprint [31].

**Distance-dependent effects of motion on functional connectivity (DM-FC).**   To determine the residual distance-dependence of subject movement, we first calculated the Euclidean distance between the centers of mass of each pair of parcels [25]. Closer parcels generally exhibit greater impact of motion on connectivity. We then correlated the distance separating each pair of parcels and the associated QC-FC correlation of the edge connecting those parcels. We report the absolute correlation values and expect to see a general trend toward zero correlation after confound regression.

**Network modularity.**   Confound regressors have the potential to remove real signals in addition to motion-related noise. In order to evaluate this possibility, we computed modularity quality, an explicit quantification of the degree to which there are structured subnetworks in a

given network - in this case the denoised connectome [4]. Modularity quality is quantified by graph community detection based on the Louvain method [36], implemented in the Brain Connectivity Toolbox [36]. If confound regression and censoring were removing real signals in addition to motion-related noise, we would expect modularity to decline. To understand the extent of correlation between modularity and motion, we computed the partial correlation between subjects' modularity values and mean framewise displacement, with age and sex as covariates.

## Supporting information

**S1 Text. Literature review and details on fMRI confounds regression.**
(PDF)

**S1 Fig. Loss of temporal degrees of freedom broke down by subgroups after removing high motion subjects.**
(TIFF)

**S2 Fig. Loss of temporal degrees of freedom broke down by subgroups before removing high motion subjects.**
(TIFF)

## Acknowledgments

Please see the original repository (https://github.com/SIMEXP/load_confounds#contributors-) for a history of initial development and contributors, and this issue (https://github.com/nilearn/nilearn/issues/2777) for a history of the integration in Nilearn and all the linked Pull Requests.

## Author Contributions

**Conceptualization:** Hao-Ting Wang, Steven L. Meisler, Hanad Sharmarke, Christopher J. Markiewicz, François Paugam, Bertrand Thirion, Pierre Bellec.

**Data curation:** Hao-Ting Wang.

**Formal analysis:** Hao-Ting Wang.

**Funding acquisition:** Pierre Bellec.

**Investigation:** Hao-Ting Wang.

**Methodology:** Hao-Ting Wang.

**Project administration:** Pierre Bellec.

**Resources:** Pierre Bellec.

**Software:** Hao-Ting Wang, Steven L. Meisler, Hanad Sharmarke, Nicolas Gensollen, François Paugam, Bertrand Thirion, Pierre Bellec.

**Supervision:** Bertrand Thirion, Pierre Bellec.

**Validation:** Hao-Ting Wang, Nicolas Gensollen, Bertrand Thirion, Pierre Bellec.

**Visualization:** Hao-Ting Wang.

**Writing – original draft:** Hao-Ting Wang, Natasha Clarke, Bertrand Thirion, Pierre Bellec.

**Writing – review & editing:** Hao-Ting Wang, Steven L. Meisler, Natasha Clarke, Christopher J. Markiewicz, Bertrand Thirion, Pierre Bellec.

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
