## [Decision Letter · Decision Letter 0]

24 Nov 2023

Dear Dr Wang,

Thank you very much for submitting your manuscript "Continuous Evaluation of Denoising Strategies in Resting-State fMRI Connectivity Using fMRIPrep and Nilearn" for consideration at PLOS Computational Biology. As with all papers reviewed by the journal, your manuscript was reviewed by members of the editorial board and by several independent reviewers. The reviewers appreciated the attention to an important topic. Based on the reviews, we are likely to accept this manuscript for publication, providing that you modify the manuscript according to the review recommendations.

Sincerely,

Catie Chang

Guest Editor

PLOS Computational Biology

Thomas Serre

Section Editor

PLOS Computational Biology

Reviewer's Responses to Questions

**Comments to the Authors:**

Reviewer #1: The manuscript “Continuous Evaluation of Denoising Strategies in Resting-State fMRI Connectivity Using fMRIPrep and Nilearn” is a very impressive body of work. The analyses employed are very comprehensive and well thought out. Moreover, the authors have gone above and beyond regarding the reproducibility of the available code by sharing an interactive notebook on the neurolibre platform. The companion notebook, means that readers will be able to understand everything about how the analysis was run. Readers can test what could have gone differently with different settings. Even more helpful, due to the completeness (from data loading to results) of the code available - readers are provided with a clear understanding of how to adapt this workflow for use on new datasets. Additionally, this manuscript is presented in concert with the authors' contributions to the heavily adopted, and open-source nilearn python package, which means that their interfaces for loading confound regressors and applying confound regression with scrubbing are expected to be highly adopted by the field. Given the richness of this analysis, a few additional questions/comments struck me while reading this work. Those are listed below.

a) The authors were correct that the loss of degrees of freedom is an important consideration when determining what denoising strategy to use. Those methods with the highest loss of degrees of freedom require more participants to be excluded from the analysis. I understand that the authors needed to exclude participants with high and moderate motion to evaluate the scrubbing.2 strategy on equal footing to the other methods. However, I was struck by how many participants needed to be excluded, particularly from specific groups (i.e. children and participants with schizophrenia). The exclusion of this many participants would result in a severe loss of power, meaning that particular group comparisons are no longer feasible. While it may not be the role of this paper to comment on thresholds for participant exclusion due to motion, I would appreciate it if the number of participants excluded were stated more clearly, perhaps by reporting on the starting n’s and % excluded in Table 3.

b) Following from comment a) I am also concerned about the possible unequal loss of degrees of freedom across subgroups of interest. For example, do strategies like compcor or scrubbing.2 result in higher losses of degrees of freedom in children than adults? Does this lead to increased downstream differences in data quality? At the very least, perhaps the authors could report the loss of degrees of freedom broken down by group. The authors would also consider metric for evaluation of strategy: loss of df within the subgroup of particular interest (i.e. patients) or the interaction between groups in loss of df. Should these measures also be reported in the workflow?

c) The evaluation of denoising strategies across versions of fMRIprep may be better summarized in a table instead of a list. In other words, rather than saying that results 1, 4, 5 or the above list were also observed in the new analysis - it might have been easier for the reader to see that concept presented in a column (with checkmarks and exes?) beside the original item.

d) There is a growing concern in the field that newer, particularly newer multi-band fMRI sequences with shorter TR, higher distortion, and smaller voxel size may have different motion concerns because 1) the higher TR allows for rhythmic breathing-related motion to be captured and 2) because the higher distortions may make it harder to register frames in the motion correction algorithm - leading to artificially reduced estimates of framewise displacement. These potential differences make me wonder if the results of this workflow would be the same if this workflow was applied to a newer dataset with multi-band acquisition parameters. I understand that the incorporation of a whole new dataset may be beyond the scope of the current investigation, but perhaps some words of caution and some mention of the need for further studies of motion benchmarks across of fMRI sequence parameters may be of merit in the discussion.

Reviewer #2: Thank you for sending me this article for review. The work describes the development and assessment of an API for reproducible denoising as well as a comparison of a broad range of common denoising strategies post fmriprep processing.

This article is excellent work and will be extremely useful for the field. Importantly it provides a comprehensive assessment of several denoising strategies and gives reasonable recommendations for the users. Further, all data and scripts are easily available through the reproducible preprint server NeuroLibre.

My only comments are as follows:

1) on p. 18 "Significance tests associated with the partial correlations were performed. P-values above the threshold of α = 0. 05 were deemed significant." - do the authors mean "above the threshold" or "below the threshold"? Greater or less than 0.05?

2) Figure 12 is somewhat confusing. I would have interpreted the "Ranking" to be both in each cell in the matrix (as denoted by the size and color of the dots, as well as (and most importantly) by row. The rows do not seem to be ranked. In particular, in the conclusion the authors recommend the simple+gsr strategy, yet this strategy does not fall on the far right of the frame. The image should either be changed or the text should make things clearer

I look forward to seeing this article in print.

Reviewer #3: General comments:

This manuscript presents a reproducible denoising benchmark for the evaluation of different research software and denoising strategies. This is a very useful tool, and the manuscript does a very good job at describing its implementation as well as summarizing the benchmark outputs for several denoising strategies, datasets, and software versions. My main concern/suggestion would be to moderate some of the paper's conclusions, as they may be extrapolating beyond the evidence provided by the benchmark results (e.g. point 7 below).

Specific comments:

1. Could you please comment on the scrubbing choice of DVARS threshold (I believe the fMRIPrep default standardized DVARS threshold is 1.5 instead of 3)?

2. Please elaborate on the details and rationale of the additional strategies “compcor6” (if I understand correctly “compcor6” represents a choice of 6 components from WM and CSF vs. “compcor” which uses a variable number of components per subject accounting for 50% of the total variance?) and “scrubbing.2” (if I understand correctly it uses a more conservative FD>0.2mm threshold definition?)

3. The chosen set of pre-defined strategies (simple, scrubbing, compcor, and ica_aroma; with or without global regression) captures well some of the common approaches used by AFNI and DPABI (e.g. doi:10.3389/fnins.2022.1073800 and doi:10.3389/fnins.2023.1069639) but it would perhaps benefit from adding another common approach used by CONN combining scrubbing and aCompCor (e.g. the default settings in the CONN toolbox when importing fMRIPrep data will use motion(n=12), WM(n=5), CSF(n=5), scrubbing(FD>.5); see doi:10.3389/fnins.2023.1092125);

4. In the 'Software implementation' section please describe explicitly the mathematical operation implemented by the denoising procedure (e.g. a linear regression of noise components, performed separately for each individual functional run).

5. Relatedly, please clarify whether scrubbing is implemented as a set of additional regressors (one per outlier timepoints) introduced in the denoising linear regression step, or if it is implemented as “censoring” (explicitly removing individual timepoints from the data, before or after other denoising steps). If the latter, I would strongly suggest to add minimally an option to use the former approach, as it would avoid the concern regarding data continuity mentioned in other parts of this manuscript (see point 7 below)

6. For the computation of QC-FC correlations, please comment on the choice of controlling by subject's age and sex. It seems counterintuitive that correlations between subject motion and functional connectivity that were caused by, for example, having larger motion in younger participants, would be explicitly disregarded when evaluating the relative success of a denoising strategy. If possible please report also FC-QC values without covariate correction.

7. The 'conclusions' section seems to assume that scrubbing is somehow incompatible with analyses that require “continuous sampling time series”. I believe this would only be true for a “censoring” approach that explicitly removes individual timepoints from the data, while a more common “regression” approach, that simply uses those individual timepoints as additional regressors either during denoising or during first-level analyses, would preserve the data continuity and avoid this concern. Given that the latter (regression-based) scrubbing approach is highly prevalent in the literature as well as across some of the most common software packages that implement denoising (e.g. AFNI, CONN, DPABI, C-PAC) I would recommend either implementing this approach if not done already (see point 5 above) or reconsidering the paper's strategic recommendations.

8. The 'conclusions' section would perhaps benefit from some additional thoughts on the potential applications and limitations of the proposed reproducible benchmark when used to help guide strategic recommendations for different denoising procedures or pipelines, as used in the paper. In particular, the results of this paper, consistent with a lot of the literature, indicate that the relative success/failure of different denoising strategies may vary strongly across different datasets (what works best for one dataset may not work for another). From that perspective, I wonder whether the current benchmark could be extended to include additional datasets, or even adapted for researchers to use on their own datasets in order to evaluate the specific merits of different denoising strategies on their own data. I would also love to hear the authors' thoughts on: a) the generalizability of the conclusions drawn from these datasets to others; and b) how this benchmark may be used not only by researchers interested in methodological advancements in denoising procedures but also perhaps by general researchers on their own datasets as part of general quality assurance procedures for functional connectivity analyses.

**Have the authors made all data and (if applicable) computational code underlying the findings in their manuscript fully available?**

Reviewer #1: Yes

Reviewer #2: Yes

Reviewer #3: Yes

PLOS authors have the option to publish the peer review history of their article (what does this mean?). If published, this will include your full peer review and any attached files.

Reviewer #1: No

Reviewer #2: No

Reviewer #3: **Yes: **Alfonso Nieto-Castanon

Figure Files:

Data Requirements:

Reproducibility:

References:

---

## [Decision Letter · Decision Letter 1]

23 Feb 2024

Dear Dr Wang,

We are pleased to inform you that your manuscript 'Continuous evaluation of denoising strategies in resting-state fMRI connectivity using fMRIPrep and Nilearn' has been provisionally accepted for publication in PLOS Computational Biology.

Best regards,

Catie Chang

Guest Editor

PLOS Computational Biology

Thomas Serre

Section Editor

PLOS Computational Biology

Reviewer's Responses to Questions

**Comments to the Authors:**

Reviewer #3: Thank you for your thoughtful responses. All my concerns were adequately addressed. Looking forward to seeing this excellent work published!

**Have the authors made all data and (if applicable) computational code underlying the findings in their manuscript fully available?**

Reviewer #3: Yes

PLOS authors have the option to publish the peer review history of their article (what does this mean?). If published, this will include your full peer review and any attached files.

Reviewer #3: **Yes: **Alfonso Nieto-Castanon

---

## [Editor Report · Acceptance letter]

7 Mar 2024

PCOMPBIOL-D-23-01130R1 

Continuous evaluation of denoising strategies in resting-state fMRI connectivity using fMRIPrep and Nilearn

Dear Dr Wang,

I am pleased to inform you that your manuscript has been formally accepted for publication in PLOS Computational Biology. Your manuscript is now with our production department and you will be notified of the publication date in due course.

With kind regards,

Judit Kozma
